# Opposing regulation of TNF responses by IFN-γ and a PGE2-cAMP axis that is apparent in rheumatoid and immune checkpoint inhibitor-induced arthritis human IL-1β+ macrophages

**Upneet K Sokhi[1†], Ruoxi Yuan[1,2†], Bikash Mishra[1,3‡], Yurii Chinenov[1,2‡], Anvita Singaraju[1,3], Karmela K Chan[4,5], Anne Bass[4,5], Richard D Bell[1,2,3], Laura Donlin[1,3,4,5], Lionel B Ivashkiv[1,3,4,5]***

[1]HSS Research Institute and David Z. Rosensweig Genomics Research Center, Hospital for Special Surgery, New York, United States; [2]Computational Biology Core, David Z. Rosensweig Genomics Research Center, Hospital for Special Surgery, New York, United States; [3]Immunology and Microbial Pathogenesis Program, Weill Cornell Medicine, New York, United States; [4]Division of Rheumatology, Department of Medicine, Hospital for Special Surgery, New York, United States; [5]Department of Medicine, Weill Cornell Medicine, New York, United States

**\*For correspondence:**
ivashkivl@hss.edu

[†]These authors contributed equally to this work

[‡]These authors also contributed equally to this work

**Competing interest:** The authors declare that no competing interests exist.

## eLife Assessment

The article contains **important** findings regarding inflammatory macrophage subsets that have theoretical and/or practical applications beyond the field of rheumatology. The authors demonstrate with **compelling** evidence the effects of PGE2 on TNF signaling. This work will be of broad interest to immunologists and cell biologists.

**Abstract** IL-1β-expressing macrophages have been described in rheumatoid arthritis (RA), immune checkpoint inhibitor-induced inflammatory arthritis (ICI-arthritis), and pancreatic cancer and proposed to be pathogenic. IL-1β+ macrophages express genes cooperatively induced by PGE2 and TNF signaling, but mechanisms that induce these cells are not known. We used an integrated transcriptomic and epigenomic analysis in primary human monocytes to study PGE2-TNF crosstalk, and how it is regulated by IFN-γ, as occurs in RA synovial macrophages. We identified a TNF + PGE2 (TP) induced gene expression signature that is enriched in IL1β+ RA and ICI-arthritis monocytic subsets, and includes genes in pathogenic IL-1, Notch and neutrophil chemokine pathways. ICI-arthritis myeloid cells mapped primarily onto four previously defined RA synovial monocytic clusters, and TP genes were expressed in a manner suggestive of a new functional monocyte subset. TP signature genes are distinct from canonical inflammatory NF-κB target genes such as *TNF*, *IL6* and *IL12B* and are activated by cooperation of PGE2-induced AP-1, CEBP and NR4A family transcription factors with TNF-induced NF-κB activity. Unexpectedly, IFN-γ suppressed induction of AP-1, CEBP and NR4A activity to ablate induction of IL-1, Notch and neutrophil chemokine genes, while promoting expression of distinct inflammatory genes such as *TNF* and T cell chemokines like CXCL10. The opposing cross-regulation of PGE2 and IFN signaling in vitro was reflected in vivo in mutually exclusive expression of TP and IFN signatures in different cell clusters in RA and ICI-arthritis monocytes. These results reveal the basis for synergistic induction of inflammatory genes by PGE2

and TNF, and a novel regulatory axis whereby IFN-γ and PGE2 oppose each other to determine the balance between two distinct TNF-induced inflammatory gene expression programs relevant for RA and ICI-arthritis.

## Introduction

Prostaglandin E2 (PGE2) is a small lipid molecule with homeostatic functions in various tissues whose production is strongly upregulated after tissue injury or inflammatory signaling. PGE2 plays a key role in early acute inflammation by increasing vascular permeability and activating mast cells, with associated tissue edema and influx of immune cells such as neutrophils (*Kawahara et al., 2015*; *Tsuge et al., 2019*). PGE2 is also a key mediator of acute pain, in part by acting directly on nociceptors (*Kawabata, 2011*). Inhibition of these PGE2-mediated functions explains the effectiveness of nonsteroidal antiinflammatory drugs (NSAIDs) that suppress PGE2 production in alleviating acute pain and tissue swelling. In contrast to activating mesenchymal cells, PGE2 has generally been considered to have primarily suppressive effects on both innate and adaptive immune cells (*Chen et al., 2012*; *Kawahara et al., 2015*; *Luan et al., 2014*; *MacKenzie et al., 2013*; *Perretti et al., 2017*; *Rodríguez et al., 2014*; *Sundberg et al., 2014*; *Yokoyama et al., 2013*). This includes inhibition of Th1 cells and IFN-γ production, and inhibition of induction of inflammatory genes such as *TNF*, *IL12B*, and *IFNB* in macrophages and dendritic cells (DCs). Such inhibition raises the possibility that the expression of these genes increases after NSAID therapy, which would contribute to lesser effectiveness and lack of disease-modifying activity of NSAIDs in chronic inflammatory conditions such as rheumatoid arthritis (RA) (*Smolen et al., 2016*).

In myeloid cells, PGE2 signals primarily via EP2 and EP4 receptors, which are G-protein-coupled receptors (GPCRs) that elevate intracellular cAMP to directly activate protein kinase A (PKA) and signaling effector EPAC, which is coupled to small GTPases Rap1/2 (*Yokoyama et al., 2013*). The immune suppressive effects of PGE2 have been attributed to PKA-mediated activation of transcription factor (TF) CREB and histone deacetylases 4 and 5 (HDAC4/5). CREB induces expression of transcriptional repressors such as *CREM* and *ATF3*, and CREB can suppress NF-κB activity in a gene-specific manner by mechanisms that are not fully understood but include competition for transcription coactivator CBP (*Altarejos and Montminy, 2011*; *Gerlo et al., 2011*). HDACs 4/5 bind to inflammatory gene loci such as *Tnf* and *Il12b* and suppress transcription (*Luan et al., 2014*). Additionally, PGE2 suppresses TLR-induced autocrine IFN responses by suppressing TLR4 trafficking and signaling via TRIF (*Perkins et al., 2018*), preventing induction of *Ifnb1* and de novo enhancers by suppressing activity of TF MEF2A (*Cilenti et al., 2021*), and attenuating induction of IRF8 target genes by suppressing IRF8 expression (*Bayerl et al., 2023*). In accord with the suppression of immune cells, activation of cAMP signaling has shown therapeutic benefit in various inflammatory conditions (*Tsuge et al., 2019*; *Yokoyama et al., 2013*). Inhibitors of phosphodiesterase 4, which increase intracellular cAMP in immune cells, have been approved by the FDA for the treatment of psoriatic arthritis, asthma, and chronic obstructive pulmonary disease (*Li et al., 2018*). However, the efficacy of these therapies may be limited by concomitant pro-inflammatory effects of cAMP signaling, such as cell-, gene-, and context-specific augmentation of NF-κB activity (*Gerlo et al., 2011*). The importance of deciphering the mechanisms by which PGE2 activates inflammatory gene expression in macrophages is highlighted by a recent study showing that cooperation between PGE2 and TNF induces IL-1β+ tumor-associated macrophages that are associated with disease progression in pancreatic ductal adenocarcinoma (*Caronni et al., 2023*).

Activated monocytes and macrophages that infiltrate inflamed synovium (joint tissues) in RA and produce inflammatory mediators and cytokines have been strongly implicated in disease pathogenesis (*Gravallese and Firestein, 2023*; *Kalliolias and Ivashkiv, 2016*; *McInnes and Schett, 2011*). High-level expression of inflammatory genes in RA synovitis has long been appreciated, and recent studies using high-dimensional single-cell profiling technologies have identified subsets of synovial monocytes/macrophages (hereafter termed MonoMacs) and highlighted a pervasive IFN signature and likely pathogenic myeloid subsets that coordinately express NF-κB target genes and interferon-stimulated genes (*Alivernini et al., 2020*; *Kuo et al., 2019*; *Lewis et al., 2019*; *Mandelin et al., 2018*; *Orange et al., 2018*; *Zhang et al., 2023*; *Zhang et al., 2019*). One RA synovial macrophage subset, termed cluster 1 in *Kuo et al., 2019*, was increased in RA relative to osteoarthritis samples, was

proposed to be pathogenic based on elevated expression of inflammatory genes, and was notable for high *IL1B* expression. IL-1*β*+ monocytes and macrophages have also been observed in inhibitor-induced inflammatory arthritis (ICI-arthritis) that occurred after anti-PD-1 therapy (*Zhou et al., 2024*). In RA, the macrophage cluster 1 gene expression signature could be modeled by coculture of monocytes with synovial fibroblasts in the presence of TNF, and a substantial fraction of the fibroblast+TNF effect was dependent upon fibroblast-produced PGE2 (*Donlin et al., 2014*; *Kuo et al., 2019*). Notably, TNF-induced and PGE2-mediated fibroblast–monocyte crosstalk cooperatively induced expression of potentially pathogenic cluster 1 genes such as *HBEGF*, *EREG*, *IL1B*, and transcription factor (TF) *STAT4*. Other studies also found that early RA synovial biopsy tissues showed enrichment of 'eicosanoid' and 'cAMP-mediated signaling' pathways (*Lewis et al., 2019*), and RNAseq of purified macrophages from synovial biopsies identified six gene modules, one of which showed enrichment for 'G-protein-coupled receptor signaling pathway' (*Mandelin et al., 2018*). Collectively, these studies establish activation of the PGE2-cAMP pathway in a likely pathogenic subset of RA synovial macrophages and suggest that crosstalk of PGE2-cAMP with TNF and possibly IFN signaling is important for the pathogenic phenotype.

Therapeutic targeting of immune checkpoint molecules such as PD-1 and CTLA4 has been highly effective in cancer therapy, but is also associated with autoimmune responses, termed immune-related adverse events (irAEs), in greater than 80% of treated patients (*Chan and Bass, 2022*). Immune checkpoint inhibition (ICI) therapy induces inflammatory arthritis in approximately 4% of patients. ICI-arthritis begins shortly after treatment and can develop into a severe and persistent polyarthritis that requires anti-inflammatory therapy and even joint replacement. Although ICI-arthritis patients are typically negative for autoantibodies diagnostic for RA, such as anti-citrullinated protein antibodies (ACPAs) or rheumatoid factor, joint pathology and synovitis can be similar to RA, with infiltration by lymphocytes and myeloid cells, pannus formation, and joint tissue destruction. Recent work has highlighted synovial and blood expansion of oligoclonal IFN-γ-expressing CD8+T cells that show evidence of activation by type I IFNs (*Kim et al., 2022*; *Wang et al., 2023*). Patients treated with combined inhibition of PD-1 and CTLA4 show elevated levels of CD4+T cells expressing IL-17 or IL-17 together with IFN-γ (also known as transient Th17 cells). The myeloid compartment includes IL-1ß-hi monocyte and macrophage subsets, and ICI-arthritis synovial fluid contains elevated levels of IL-1β, IL-6, IFN-γ, and IL-17A (*Kim et al., 2022*; *Zhou et al., 2024*). These findings raise the possibility that crosstalk among these infiltrating immune cells and their secreted products could generate myeloid cell subsets similar to those in RA.

We wished to understand the full spectrum of pathogenic inflammatory genes induced by cooperation between PGE2 and TNF signaling to gain insight into mechanisms underlying signaling crosstalk and address the important question of how PGE2-TNF crosstalk is modulated by IFN-γ, as occurs in RA synovial macrophages. We performed an integrated transcriptomic and epigenomic analysis of gene expression and chromatin accessibility using primary human monocytes, which correspond directly to cells that migrate into inflamed synovium and provide an opportunity to model in vivo activation. We identified a 'TNF+PGE2' (TP) gene expression signature that is enriched in RA synovial monocytic subsets and includes genes in pathogenic IL-1, Notch, and neutrophil chemokine pathways. A similar gene expression signature was also apparent in ICI-arthritis myeloid subsets that showed similarity to pathogenic RA monocytic subtypes. Expression of the TP signature was driven by cooperation of PGE2-induced AP-1, CEBP, and NR4A family TFs with TNF-induced NF-κB activity. IFN-γ suppressed induction of AP-1, CEBP, and NR4A activity and suppressed IL-1, Notch, and neutrophil chemokine genes, while promoting expression of distinct inflammatory genes and T cell chemokines. Mutually exclusive expression of TP and IFN signatures in distinct RA and ICI-arthritis myeloid cell subsets supported similar opposing cross-regulation amongst these pathways in vivo. These results reveal a novel regulatory axis whereby IFN-γ and PGE2 oppose each other to determine the balance between two distinct TNF-induced inflammatory gene expression programs relevant for the pathogenesis of RA and ICI-arthritis.

## Results

### PGE2 augments expression of a large subset of TNF-inducible genes

We used RNAseq to perform a transcriptomic analysis of the effects of PGE2 on the TNF response in monocytes. Primary human monocytes were stimulated with PGE2 (280 nM) and/or TNF (20 ng/ml) and harvested 3 or 24 hr after stimulation for RNAseq analysis; the experimental design is depicted in *Figure 1A*. As expected, co-stimulation of monocytes with PGE2+TNF induced expression of genes enriched in inflammatory and NF-κB pathways, largely reflecting a TNF response (*Figure 1B*). Clustering of genes upregulated after 3 hr of stimulation (n=747; fold change >2, false discovery rate [FDR] < 0.05) based upon pattern of expression confirmed that PGE2 suppresses expression of a subset of TNF-inducible genes including *TNF* itself and interferon-stimulated genes *CXCL9*, *CXCL10,* and *CXCL11* (*Figure 1C*, group III, n=86 suppressed genes). Surprisingly, PGE2 augmented TNF-mediated induction of a substantially larger number of genes (*Figure 1C*, groups I and V, 395 co-stimulated genes). Pathway analysis revealed that genes costimulated by PGE2+TNF were similarly enriched in inflammatory pathways (e.g., response to lipopolysaccharide, neutrophil migration), although the enriched pathways were partially distinct (*Figure 1D*). We also observed two groups of genes that were induced by PGE2 alone, which included canonical cAMP signaling and CREB target genes (*Zhang et al., 2005*) such as *HBEGF* and *CREM* (*Figure 1C*, groups II and IV). RT-qPCR analysis confirmed strong costimulation of *IL1B*, *IL1A*, and *CSF3*, induction of *HBEGF* by PGE2 alone, and suppression of TNF-induced expression of *TNF* and *CXCL10* by PGE2 (*Figure 1E*). Similar results on the regulation of TNF-mediated gene expression by PGE2 were observed using mouse bone marrow-derived macrophages (BMDMs) (*Figure 1—figure supplement 1*). Subsets of inducible genes showed distinct kinetics of induction with transient expression at 3 hr versus sustained expression over the 24 hr stimulation period. Costimulation of STAT4 was confirmed at the protein level (*Figure 1F*). Analysis of RNAseq performed at the 24 hr time point to determine the effects of PGE2 on the late phase TNF response showed similar results and additionally revealed that PGE2 inhibits TNF-induced expression of cholesterol pathway genes, which we had previously shown to be induced by TNF with delayed kinetics and to play a role in promoting IFN responses (*Kusnadi et al., 2019*; *Figure 1—figure supplement 2*). Collectively, these results indicate that PGE2 has a dichotomous effect on TNF-induced gene expression, superinducing select key inflammatory genes such as *IL1B*, while suppressing expression of distinct inflammatory genes including *TNF* and interferon-stimulated genes in line with previous reports (*Chen et al., 2012*; *Cilenti et al., 2021*; *Gerlo et al., 2011*; *Kawahara et al., 2015*; *Luan et al., 2014*; *MacKenzie et al., 2013*; *Sundberg et al., 2014*; *Tsuge et al., 2019*).

### The (TNF+PGE2) signature is expressed in similar RA and ICI-arthritis synovial myeloid cell subsets

The above-described results were similar to results we had previously observed using TNF-stimulated monocyte–synovial fibroblast cocultures (*Donlin et al., 2014*; *Kuo et al., 2019*), except that PGE2-inducible genes such as *HBEGF* were strongly induced by PGE2 alone, and expression was not further increased by TNF (*Figure 1C*, groups II and IV). These differences may reflect delayed kinetics of induction of fibroblast-derived PGE2 in cocultures or from the modulating activity of various factors and cytokines produced by activated fibroblasts in these cultures. To assess the extent to which TNF-stimulated monocyte-fibroblast cross talk could be recapitulated by stimulation of monocytes with purified PGE2 and TNF, we compared the current (TNF+PGE2)-costimulated gene set with the TNF/fibroblast-costimulated gene set previously reported (*Donlin et al., 2014*; *Kuo et al., 2019*). Strikingly, (TNF+PGE2)-costimulated genes recapitulated 61% of fibroblast-augmented TNF-inducible genes (*Figure 2A*, hypergeometric p=8.36e-128), which is in accord with our previous conclusions that PGE2 mediates a substantial fraction of the fibroblast effect on monocytes (*Kuo et al., 2019*). Since TNF-stimulated fibroblast-cocultured monocytes modeled the phenotype of RA synovial macrophage cluster 1 (C1, also termed HBEGF+ or IL1B+ macrophages) (*Kuo et al., 2019*; *Zhang et al., 2023*; *Zhang et al., 2019*), we next tested whether (TNF+PGE2)-stimulated monocytes expressed the defining genes of cluster 1 (128 genes, as defined in *Kuo et al., 2019*). Strikingly, (TNF+PGE2)-costimulated genes (*Figure 2B*, orange+yellow) highly significantly mimicked the RA C1 phenotype (p<10e-9 by Monte Carlo simulation) and recapitulated the C1 phenotype more completely than TNF/fibroblast-costimulated genes (*Figure 2B*, orange+red). The (TNF+PGE2)-costimulated genes were parts of pathogenic gene modules related to IL-1-NF-κB and Jak-STAT signaling, neutrophil

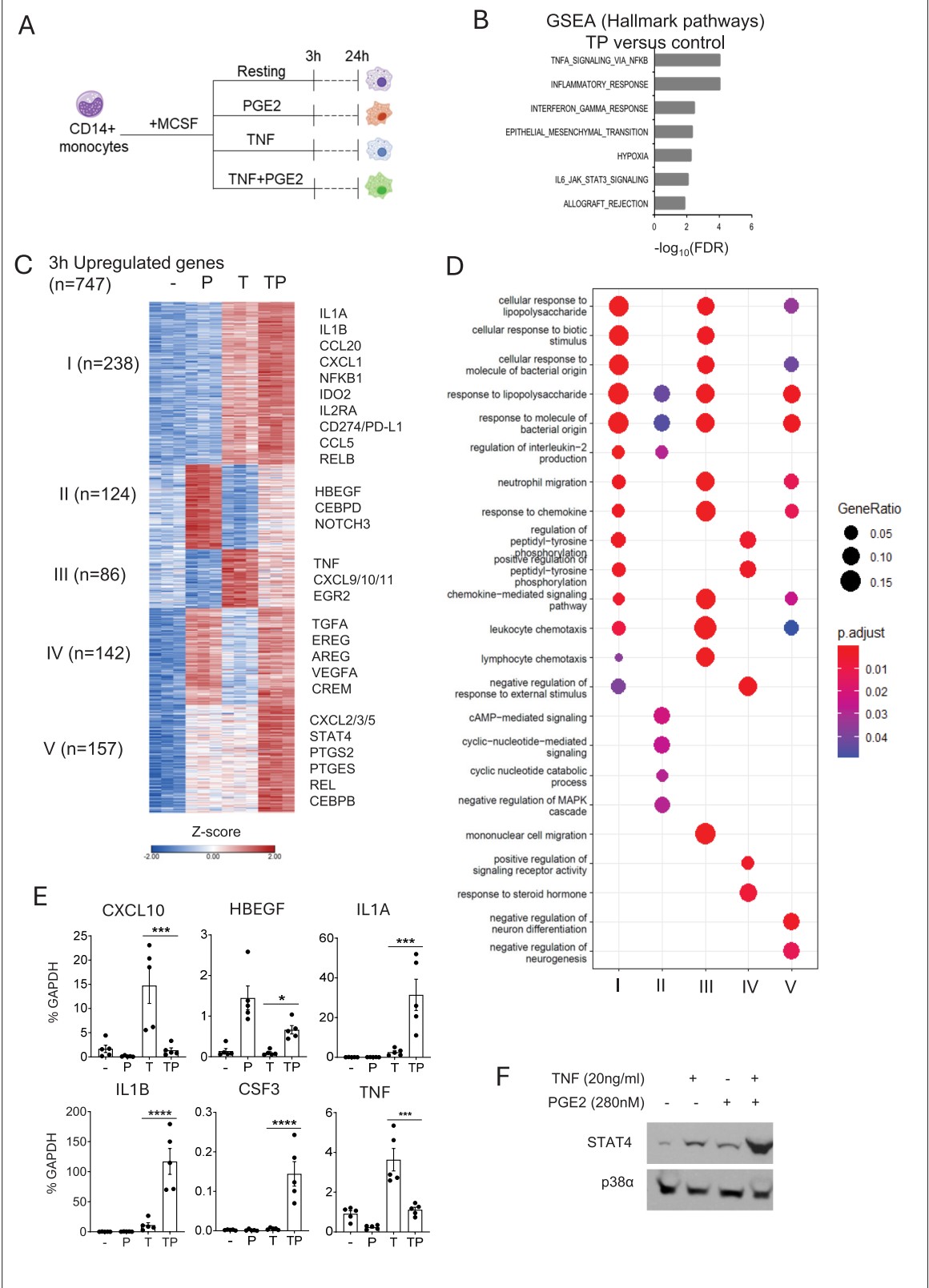

**Figure 1.** Cooperative induction of a distinct subset of inflammatory genes by PGE2 and TNF. (**A**) Experimental design. Primary human monocytes were stimulated with PGE2 (280 nM) and/or TNF (20 ng/ml) and harvested 3 or 24 hr after stimulation for RNAseq analysis. n=3 independent blood donors. (**B**) Gene set enrichment analysis of genes induced >2-fold by PGE2+TNF (TP) (false discovery rate [FDR] < 0.05). (**C**) K-means clustering of differentially upregulated genes in any pairwise comparison relative to resting control (>2-fold induction, FDR < 0.05). 3 hr time point. k=5. (**D**) Pathway

*Figure 1 continued on next page*

*Figure 1 continued*

analysis of gene clusters in panel (**C**). (**E**) qPCR analysis of gene expression in an additional five blood donors. Mean ± SEM. Statistical significance was assessed using one-way ANOVA and Sidak's test for multiple comparisons (*p<0.05, **p<0.01, ***p<0.001, ****p<0.0001). (**F**). Western blot analysis. Representative blot out of four independent experiments.

The online version of this article includes the following source data and figure supplement(s) for figure 1:

**Source data 1.** Original western blot images.

**Source data 2.** PDF of original western blot images with relevant bands and treatments indicated.

**Figure supplement 1.** Regulation of gene expression by PGE2 and TNF in mouse bone marrow-derived macrophages.

**Figure supplement 2.** Analysis of differentially regulated genes at the 24 hr time point.

chemotaxis, and the Notch pathway that has been recently implicated in RA pathogenesis (***Wei et al., 2020***; ***Zack et al., 2024***; ***Figure 2C***, blue font = C1 genes). Induction of neutrophil chemokines such as CXCL1/2/3/5 that recruit neutrophils to inflamed tissues complements induction of CSF3 that mobilizes neutrophils from the bone marrow. Additional examples of induction of RA macrophage C1 genes by PGE2 and TNF are depicted in ***Figure 2D***. These results link (TNF+PGE2)-costimulated macrophages to a pathogenic RA macrophage phenotype and reveal that, instead of global suppressive effects in human MonoMacs, PGE2 promotes induction of genes important in various pathogenic inflammatory pathways.

We wished to test whether macrophage subsets expressing TP-costimulated genes could be detected in other types of inflammatory arthritis and thus investigated synovial macrophages from patients with ICI-arthritis. While immune checkpoint inhibitors are effective in the treatment of malignancies, ICI can lead to autoimmune side effects including inflammatory arthritis (***Dougan et al., 2021***). Approximately 4% of ICI-treated patients can develop autoimmune inflammatory arthritis that persists despite cessation of ICI therapy (***Braaten et al., 2020***), and understanding mechanisms and pathways that activate immune cells in ICI-arthritis may yield insights into the pathogenesis of spontaneous autoimmune arthritides (***Cappelli et al., 2020***). We performed single-cell RNA sequencing analysis of mononuclear cells FACS-sorted from synovial fluids of five ICI-arthritis patients and from two synovial tissues from one ICI-arthritis patient who underwent bilateral knee replacements. These patients had received anti-PD1 antibodies as monotherapy or in combination with anti-CTLA4 therapy (clinical characteristics of patients are presented in ***Supplementary file 1***). Following subclustering of monocytes and macrophages, we defined eight clusters from 14,110 cells (***Figure 2E***). Cluster 4, accounting for 15% of cells, selectively showed elevated expression of (TNF+PGE2)-costimulated genes and known PGE2 targets, including *HBEGF, IL1B, CXCL2/3/8, CREM,* and *PLAUR* (***Figure 2F***, genes marked in red). Cluster 4 macrophages are distinct from those in the other clusters, such as clusters 0 and 1 that express interferon-stimulated and different inflammatory genes. These results suggest that costimulation of gene expression by TNF and PGE2 can occur in ICI-arthritis.

We next wished to analyze the relationship of ICI-arthritis myeloid cells to previously defined RA pathogenic synovial MonoMac subsets. To this end, we retrieved the RA myeloid cell scRNAseq dataset from ***Zhang et al., 2023***. We utilized the same 15 myeloid cell clusters defined in the original study (***Zhang et al., 2023***) and generated an essentially similar uniform manifold approximation and projection (UMAP) plot that displayed the myeloid clusters including proposed pathogenic IL1B+ FCN1+ HBEGF+ (termed 'inflammatory'), STAT1+ CXCL10 + (express high IFN signature), and SPP1+ (express 'wound healing' phenotype) monocytic subsets (***Figure 3A***). Reference mapping (***Hao et al., 2024***; ***Lotfollahi et al., 2024***) showed that the majority of ICI-arthritis myeloid cells mapped onto four RA myeloid clusters: IL1B + FCN1 + HBEGF+, STAT1+ CXCL10+, SPP1+, and MERTK+ HBEGF+ clusters (***Figure 3B***). The correspondence of ICI-arthritis cells to these specific RA clusters was corroborated by an independent computational approach involving integration, reclustering, and within-cluster dataset overlap analysis of the two datasets (not shown). These results support the similarity of certain ICI-arthritis and RA myeloid cell types, but this interpretation is subject to several caveats (see 'Discussion'). We then investigated the expression of TP gene sets in RA and ICI-arthritis myeloid clusters by calculating gene set activity scores in each cell using the AUCell package (***Aibar et al., 2017***) as described in 'Materials and methods', and visualizing gene expression on UMAP plots. The gene sets analyzed were the most synergistically induced TP genes from clusters I and V (as defined in ***Figure 1C***), and TP genes with proposed pathogenic functions similar to the gene groups shown

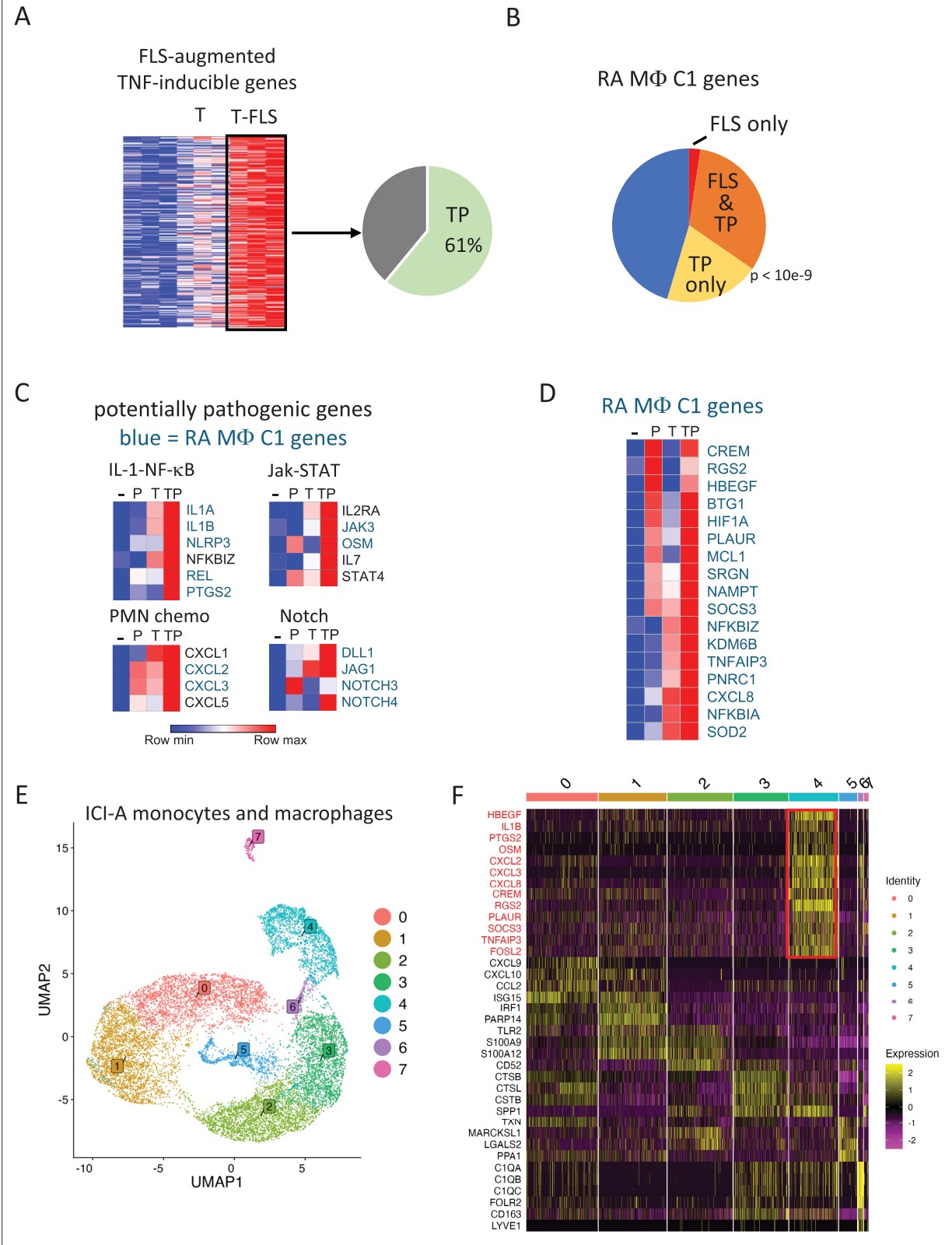

**Figure 2.** PGE2 and TNF costimulation model aspects of the RA and ICI-arthritis synovial macrophage phenotype. (**A**) TP costimulation recapitulates expression of 61% of the genes whose induction by TNF was augmented by coculture with synovial fibroblasts (also termed fibroblast-like synoviocytes, FLS). FLS-augmented TNF-inducible genes from data in *Donlin et al., 2014*; *Kuo et al., 2019* were compared to the TP-induced genes at the 24 hr time point in *Figure 1—figure supplement 2*, fold-change > 2, false discovery rate [FDR] < 0.05. The green area indicates the extent of overlap,

*Figure 2 continued on next page*

*Figure 2 continued*

hypergeometric p=8.36e-128. (**B**) Recapitulation of the RA synovial macrophage cluster 1 phenotype by TP-costimulated genes. The defining 128 genes of the C1 phenotype were overlapped with TP-costimulated genes and TNF/FLS-costimulated genes. TP-costimulated genes = orange + yellow = 52% of C1 defining genes (p<10e-9 by Monte Carlo simulation). TNF/FLS-costimulated genes = orange + red=34% of C1 defining genes. (**C**) Heatmaps depicting expression of genes in pathogenic pathways, based on RNAseq data shown in *Figure 1*. Blue font = genes expressed in C1 RA macrophages. (**D**) Heatmap depicting regulation of representative genes that are expressed in RA C1 macrophages by P, T, or TP. (**E**) UMAP visualization of monocyte and macrophage clusters based on scRNAseq of 14,110 macrophages and monocytes from 5 synovial fluids and 2 synovial tissues of ICI-arthritis patients. (**F**) Heatmap showing expression of key genes for the eight clusters identified in panel (**E**). TNF+PGE2 signature genes are shown in red.

in *Figure 2C*; this gene list was comprised of *IL1A, IL1B, NLRP3, CD274, RELB, CXCL1/2/3/5, STAT4, PTGS2, PTGES, REL, CEBPB, IL2RA, DLL1, JAG1, NOTCH4, CREM, HBEGF, PLAUR, NR4A1/2/3*, and *KDM6B*. Analysis of RA myeloid cells showed that expression of the TP genes aligned most closely with the IL1B+ FCN1+ HBEGF+ cluster, although gene expression extended into subsets of adjacent SPP1+, MERTK+ LYVE1-, and DC2 clusters (*Figure 3C*). Notably, the TP signature was absent from the IFN signature-expressing STAT1+ CXCL10+ cells. We then analyzed the expression of TP genes in ICI-arthritis myeloid cells (projected onto RA cell clusters as shown in *Figure 3B*). TP gene expression in ICI-arthritis cells was apparent in the same general and contiguous area of the projected UMAP and spanned the same clusters as expression in RA cells, except TP genes were expressed in a smaller fraction of cells in some of the RA-defined cell clusters (*Figure 3D*). Expression of TP genes in a central and contiguous region of the UMAP is suggestive of a functional myeloid cell subset associated with TP gene-related activity that is partially distinct from clusters previously identified in *Zhang et al., 2023*. Collectively, these results highlight the correspondence of ICI-arthritis myeloid cells to RA monocytic cells in defined pathogenic RA cell clusters and identify myeloid cell subsets that show evidence of co-stimulation by TNF and PGE2 in RA and ICI-arthritis.

## cAMP signaling has dichotomous suppressive and augmenting effects on the TNF-induced inflammatory response

We wished to test whether both the pro- and anti-inflammatory effects of PGE2 on TNF-induced gene expression were mediated by cAMP signaling. We addressed this question using selective agonists of PGE2 receptors EP2 and EP4 that signal predominantly via cAMP, and also cell membrane-permeable dibutyryl-cAMP that directly activates this pathway. Selective agonists of EP2 and EP4 augmented TNF-induced expression of *IL1B, IL1A, STAT4,* and *CXCL1*, while suppressing induction of the ISG *CXCL10* (*Figure 4A*). Dibutyryl-cAMP showed similar results (*Figure 4B*). These results implicate cAMP signaling in both anti- and pro-inflammatory effects in this experimental system.

## PGE2 effects on TNF-induced changes in chromatin accessibility

We then performed ATACseq analysis of chromatin accessibility (*Figure 5—figure supplement 1A*) to gain insight into how PGE2 costimulates TNF-induced gene expression. In accord with different signaling pathways activated by these factors, TNF and PGE2 induced mostly distinct ATACseq peaks (*Figure 5A*); indeed, out of 15,190 induced ATACseq peaks (FDR < 0.05, fold induction >2), only 1547 peaks (10.2%) were commonly induced by individual stimulation by TNF or PGE2 (*Figure 5A*, columns 4+7). As expected, TF binding motifs most significantly enriched under TNF-induced ATACseq peaks corresponded to AP-1 and NF-κB binding sites; additional enriched motifs included RUNX and IRF1 sites, the latter of which is in accord with the role of IRF1 in the delayed TNF-induced autocrine IFN response (*Yarilina et al., 2008*; *Figure 5B*, left). Similar to TNF, PGE2 induced peaks enriched for AP-1 motifs and additionally induced peaks with highly significant enrichment of CEBPD and NR4A1 motifs (*Figure 5B*, middle), which is in accord with PGE2-mediated induction of expression of TFs belonging to these three TF families (*Altarejos and Montminy, 2011*; *Zhang et al., 2005*). In line with the largely distinct open chromatin regions (OCRs) induced by TNF and PGE2, GREAT analysis of genes associated with these peaks showed enrichment of genes in largely distinct pathways, with TNF-induced peaks associated with genes in immune and cytokine pathways, whereas PGE2-induced peaks were associated with genes in cell matrix interactions and chemotaxis (*Figure 5—figure supplement 1B and C*).

To examine how these disparate factors could interact to costimulate gene expression, we extended our analysis to peaks that were co-induced by both TNF and PGE2 (*Figure 5A*, column 4; n=1531)

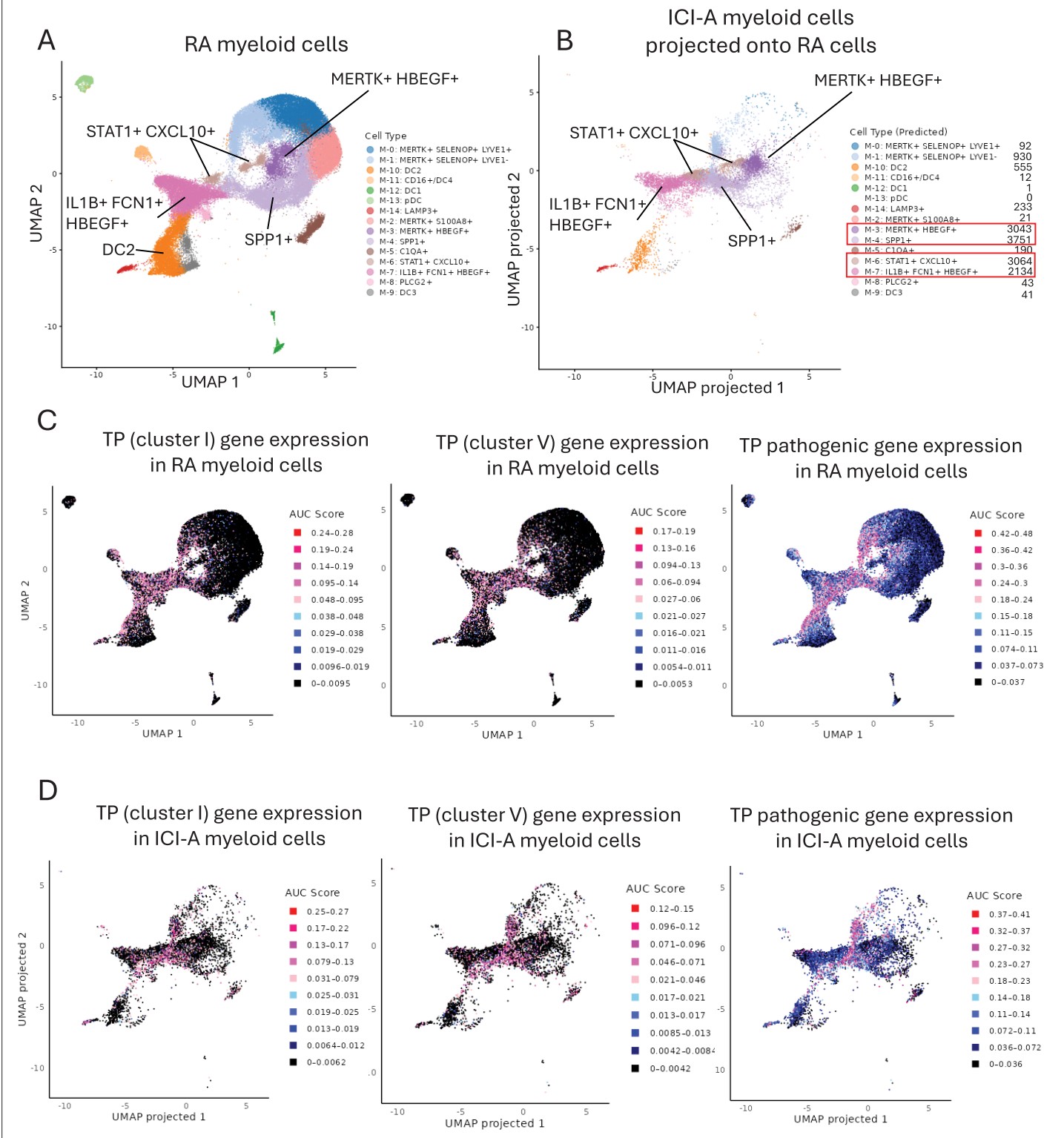

**Figure 3.** Expression of the TP gene signature in select subsets of RA myeloid cells and in ICI-arthritis myeloid cells that map onto RA cell clusters. (**A**) UMAP plot of the myeloid scRNAseq dataset from *Zhang et al., 2023* (syn52297840). The dimensionality reduction was successfully recapitulated using the uwot model provided in syn52297840. (**B**) Reference mapping of ICI-arthritis myeloid cells onto the predefined RA myeloid cell clusters. ICI-arthritis myeloid cells were projected onto the RA reference UMAP space using Seurat's MapQuery function, based on the same uwot model used in **A**. (**C**) Expression of TP gene sets in RA myeloid cells from (**A**). (**D**) Expression of TP gene sets in ICI-arthritis myeloid cells that have been mapped onto RA clusters as (**B**). In (**C**) and (**D**) gene set activity scores for individual cells were calculated using the AUCell package and area under the curve (AUC) scores are shown. Pink and red dots represent higher levels of gene expression that passed the threshold set by the AUCell algorithm.

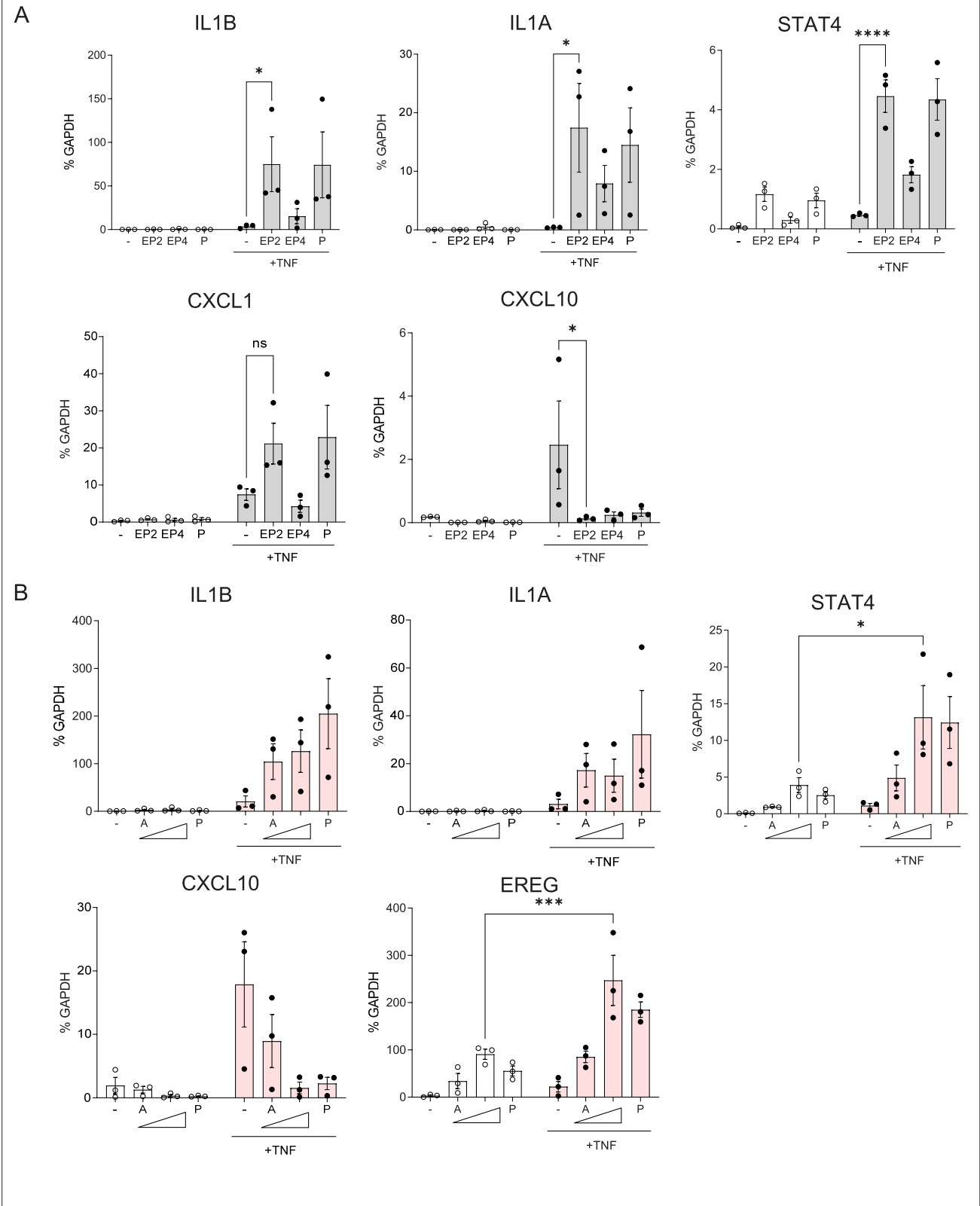

**Figure 4.** cAMP signaling has dichotomous suppressive and augmenting effects on the TNF-induced inflammatory response. (**A**) RT-qPCR analysis of gene expression in primary human monocytes stimulated with TNF and selective agonists of PGE2 receptors EP2 and EP4 that signal predominantly via cAMP. EP2 agonist = butaprost (10 µM); EP4 agonist = CAY10598 (10 µM). n=3. (**B**) RT-qPCR analysis of gene expression in primary human monocytes stimulated with TNF and increasing concentrations of cAMP analog dibutyryl cAMP (10 and 100 µM; labeled A). n=3. Mean ± SEM. Statistical significance was assessed using two-way ANOVA and Sidak's test for multiple comparisons (*$p < 0.05$, **$p < 0.01$, ***$p < 0.001$, ****$p < 0.0001$).

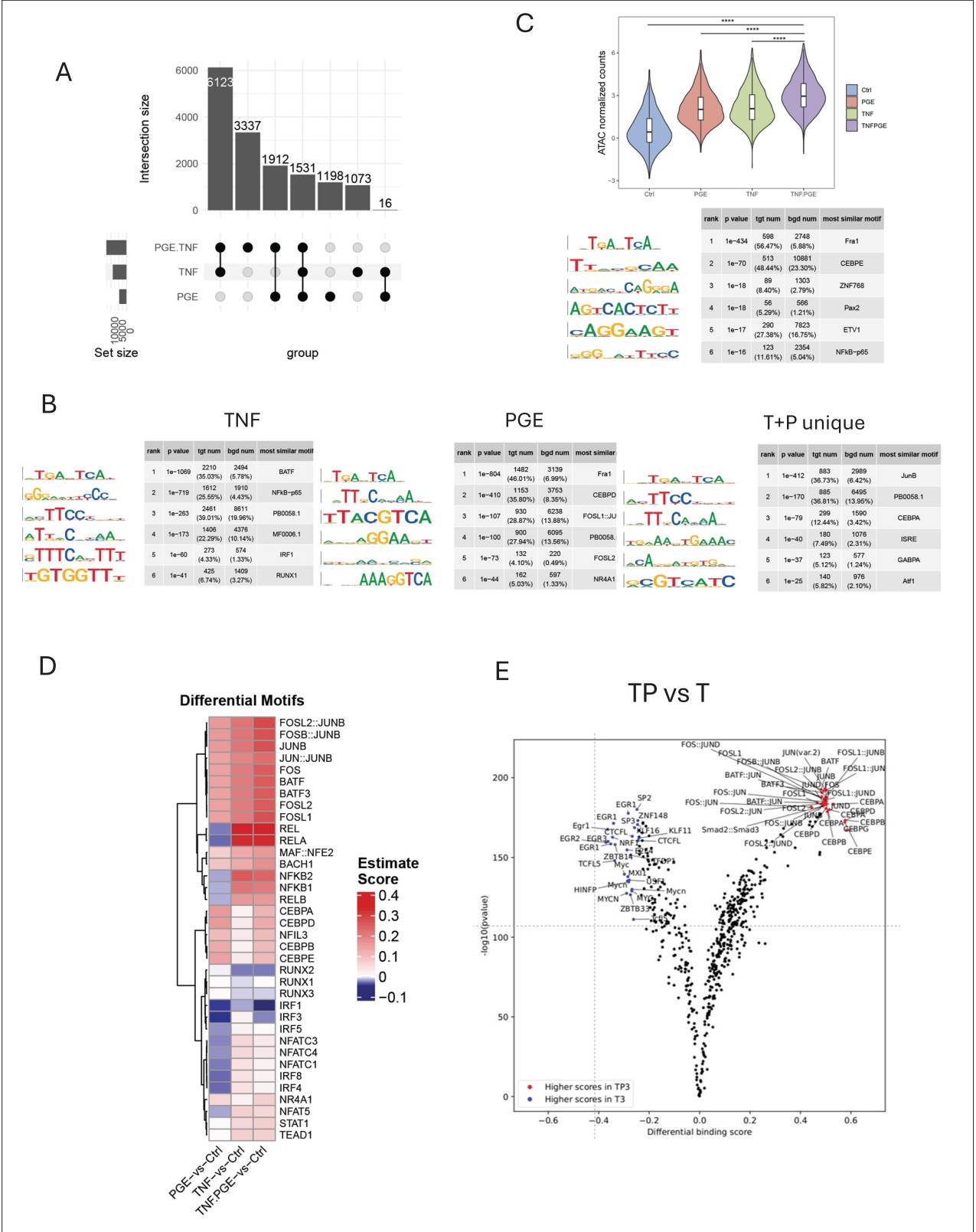

**Figure 5.** PGE2 effects on TNF-induced changes in chromatin accessibility. (**A–E**). Analysis of ATACseq data obtained using monocytes from 3 independent donors. (**A**) UPSET plot of differentially upregulated ATACseq peaks in any pairwise comparison relative to resting control (>2-fold induction, false discovery rate [FDR] < 0.05). (**B**) De novo motif analysis using HOMER of ATACseq peaks induced by TNF (left panel), PGE2 (middle panel), or uniquely induced only under conditions of TP costimulation (right panel). (**C**) Upper. Violin plots showing normalized counts of ATACseq peaks

*Figure 5 continued on next page*

*Figure 5 continued*

induced by both PGE2 and TNF. ****p<0.0001 by Wilcoxon rank sum test with Holm's correction for multiple comparisons. Lower. HOMER de novo motif analysis of the peaks in the upper panel. (**D**) Heatmap of the differential TF activity scores derived from ChromVAR analysis of ATACseq data for P, T, or TP-treated monocytes compared to resting control. (**E**) Volcano plot of differential binding analysis of ATACseq peaks between the TP and T conditions using TOBIAS.

The online version of this article includes the following figure supplement(s) for figure 5:

**Figure supplement 1.** ATACseq analysis of PGE2 and TNF-stimulated monocytes.

and peaks that were induced only under conditions of costimulation (***Figure 5B***, column 2, n=3337; termed T+P unique). TNF and PGE2 cooperated to increase chromatin accessibility at co-induced peaks (***Figure 5C***, upper), and the most significantly enriched motifs were AP-1 (Fra1, also known as FOSL1) and CEBPE (***Figure 5C***, lower). These co-induced peaks were associated with immune and inflammatory response genes (***Figure 5—figure supplement 1D***). The peaks that were uniquely induced by PGE2+TNF were most significantly enriched for AP-1, CEBP, and ISRE/IRF motifs, and also for GAPBA, KLF7, and PBX1 binding sites (***Figure 5B***, right, and data not shown), and were associated with immune response and cytokine genes (***Figure 5—figure supplement 1E***). Collectively, the results suggest that TNF and PGE2 signals converge on gene regulatory elements associated with inflammatory genes to increase chromatin accessibility and induce de novo enhancers and highlight a potential role for AP-1 and CEBP TFs in costimulating TNF-induced gene expression.

We then used ChromVAR (***Schep et al., 2017***) to assess induction of TF activity by PGE2, TNF, or their combination (***Figure 5D***). This analysis reinforced the notion that, relative to TNF alone, the PGE2+TNF costimulation condition was characterized by increased AP-1 and CEBP activity, and also NR4A1 activity. We then performed a TOBIAS analysis that uses footprinting to measure actual occupancy of TF motifs within ATACseq peaks (***Bentsen et al., 2020***). TNF stimulation induced the most significant occupancy of NF-κB and AP-1 motifs, whereas PGE2 stimulation induced AP-1 and CEBP occupancy (***Figure 5—figure supplement 1F***). A direct pairwise comparison of T+P to T revealed the most significant increases in AP-1 and CEBP occupancy (***Figure 5E***). Overall, all three complementary approaches show that, relative to TNF alone, PGE2 costimulation boosts AP-1 and adds CEBP activation, and these factors can cooperate to increase chromatin accessibility at commonly targeted sites and to induce de novo enhancer formation.

## IFN-γ opposes the effects of PGE2 on TNF-induced gene expression

In inflamed RA synovium, macrophages that express a 'TP signature' are also exposed to IFN-γ (***Kuo et al., 2019***; ***Orange et al., 2018***; ***Zhang et al., 2023***; ***Zhang et al., 2021***; ***Zhang et al., 2019***), which not only induces ISGs but also augments TLR- and TNF-induced expression of various inflammatory NF-κB target genes including *TNF*, *IL6*, and *IL12* family members (***Mishra and Ivashkiv, 2024***). We next examined the effects of IFN-γ on the expression of (TNF+PGE2)-costimulated inflammatory genes. Surprisingly, IFN-γ essentially completely abolished induction of inflammatory genes *IL1B* and *CXCL2,* and the Notch target gene *HEY1* (***Figure 6A***); as a specificity control, induction of *STAT4* was minimally affected. We next tested whether IFN-γ broadly suppressed (T+P)-costimulated genes by analyzing the effects of IFN-γ on induction of subsets of these genes in our RNAseq dataset, as defined in ***Figure 6—figure supplement 1A***. Interestingly, IFN-γ suppressed expression of only approximately 27% of genes that were most clearly synergistically induced by (T+P) (***Figure 6B***; this group included IL-1 pathway genes *NLRP3* and *PTGS2*). Similarly, IFN-γ suppressed induction of only a subset of additional genes that were costimulated by (T+P) (***Figure 6—figure supplement 1B and C***; these suppressed genes included IL-1 family members *IL1B* and *IL1A*, neutrophil chemokine genes *CXCL1/2/3/5/8,* and canonical cAMP signaling targets such as *EREG* and *NR4A1/3*). Thus, IFN-γ suppressed expression of genes in the key RA pathogenic pathways described in ***Figure 2***. This opposition between IFN-γ and PGE2 was also apparent in the regulation of a subset of TNF-induced genes that were suppressed by PGE2, whose expression was substantially higher under IFN-γ-stimulated conditions (***Figure 6C and D***). Genes more highly expressed under IFN-γ-stimulated conditions included inflammatory genes such as *TNF* and ISGs such as *CXCL10*. Collectively, the results support a biology whereby PGE2 and IFN-γ at least in part oppose each other by regulating TNF-induced gene expression in different directions and suggest that IFN-γ inhibits select inflammatory pathways while promoting others.

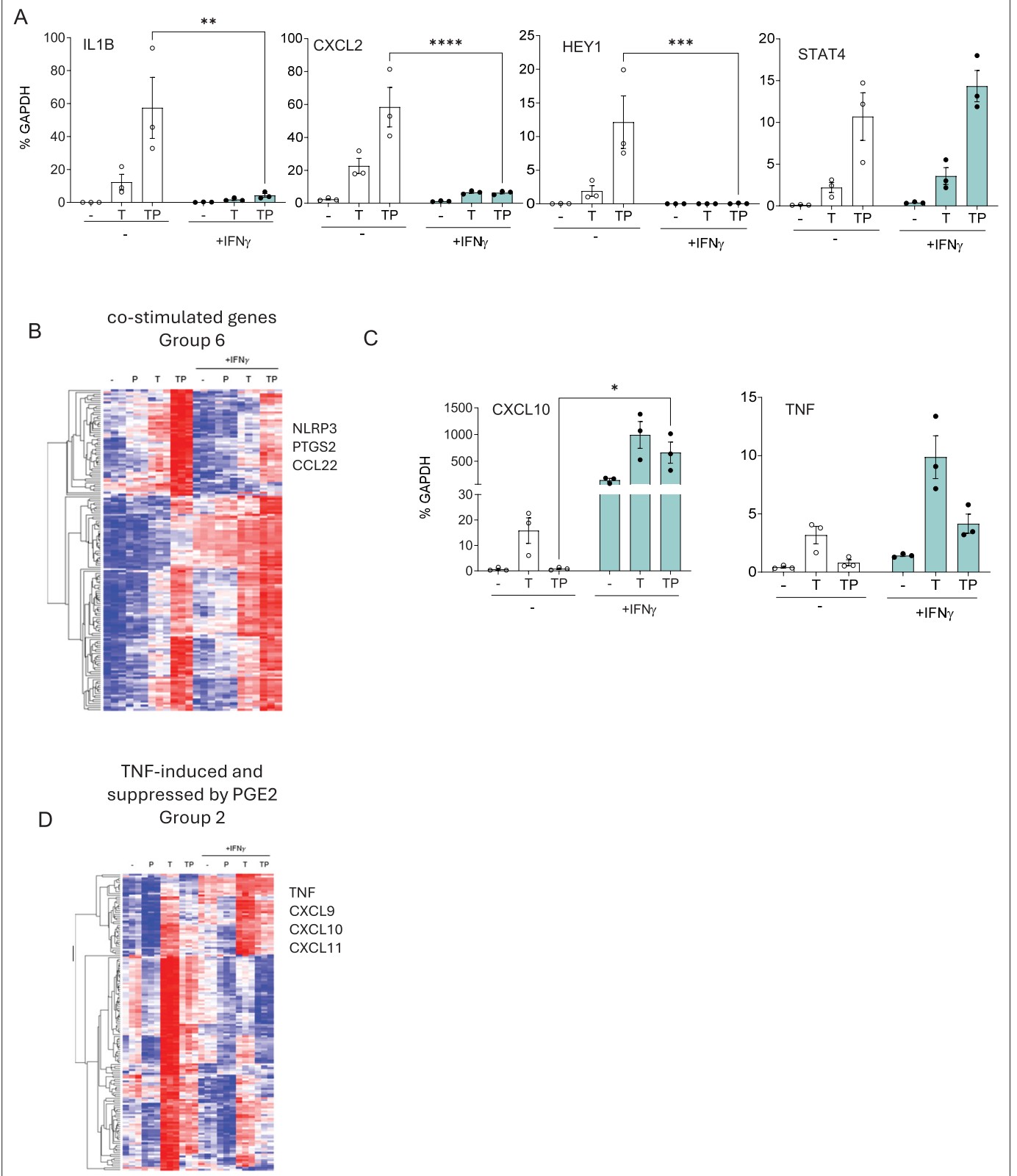

**Figure 6.** IFN-γ opposes the effects of PGE2 on TNF-induced gene expression. (**A, C**) RT-qPCR analysis of gene expression in primary human monocytes that were primed overnight with IFN-γ (100 U/ml) and then stimulated for 3 hr with P, T, or TP as in *Figure 1*. n=3. Mean ± SEM. Statistical significance was assessed using two-way ANOVA with Sidak's test for multiple comparisons (*p<0.05, **p<0.01, ***p<0.001, ****p<0.0001). (**B, D**) Gene groups defined in *Figure 6—figure supplement 1* based on pattern of expression in RNAseq data (n=3) were subjected to hierarchical clustering.

*Figure 6 continued on next page*

*Figure 6 continued*

The online version of this article includes the following figure supplement(s) for figure 6:

**Figure supplement 1.** IFN-γ opposes the effects of PGE2 on TNF-induced gene expression.

To obtain a broader view of the effects of IFN-γ on the entire TP-induced gene response, we performed a general linear model with between-treatment interaction contrast using our RNAseq dataset (*Figure 7A and B*). The conclusion that IFN-γ opposes a large component of the (TNF+PGE2) response was supported by the identification of three positive interaction clusters (ICs) comprised of genes regulated in opposing directions when IFN-γ was added to (T+P) (*Figure 7A*, ICs IC2, IC3, and IC6 where red and blue lines diverge). Genes whose suppression by TP was reversed by IFN-γ (IC2) were enriched in IFN pathways and regulated by IRF TFs, implicating IRFs in opposing negative regulation of gene expression by (T+P). In contrast, genes that were induced by (T+P) but strongly suppressed when IFN-γ was added to the TP condition were enriched in Myc targets (IC6), suggesting counter-regulation of metabolic pathways in these terminally differentiated and nonproliferating cells. Interestingly, IFN-γ minimally affected the (T+P)-induced NF-κB response (IC4). Genes that comprise the two negative ICs are upregulated (IC1) or downregulated (IC5) by IFN-γ or (T+P), but in combination IFN-γ did not provide an additional effect.

To explore the regulatory drivers of ICs, we selected all interacting differentially expressed TFs (|logFC|>1 in the interaction contrast) and built a protein association network using stringDB (*Figure 7C*). The size of a node reflects the number of unique functional connections between a node and other nodes in the network and edges connect associated nodes. Notably, TFs were grouped together with their targets in several ICs, for example, IC2 contains IRF2 and IRF8 and their targets; IC4 contains NF-κB1,2 and RELB and their targets; IC6 contains Myc and its targets (*Figure 7B and C*). Although RNA levels of Jun are regulated only weakly, its ultimate activity and binding site repertoire are affected by relative levels of dimerization partners and interacting proteins (*Chang et al., 2018*). For example, BATF2 (IC2) and BATF (IC6) were up- and down-regulated by the combination of IFN-γ and (T+P) relative to (T+P), which potentially affects relative abundance of JUN/BATF/IRF ternary complexes. Similarly, CEBPA, CEBPE (IC3), and MYC (IC6) are regulated in a reciprocal fashion: IFN-γ relieves a (T+P) inhibition of CEBPA and CEBPE yet also concomitantly decreases MYC mRNA levels. A recent report describing negative regulation of Myc by CEBPA/E (*Theilgaard-Mönch et al., 2022*) suggests that a similar mechanism is involved in IFN-γ-dependent inhibition of Myc and its targets. Overall, the interaction analysis indicates that IFN-γ does not alter the core NF-κB response (which is activated primarily by TNF) and suggests that instead IFN-γ regulates the PGE2 response, at least in part by modulating PGE2-induced AP-1 and CEBP factors.

## IFN-γ inhibits PGE2-induced gene expression and chromatin accessibility

We tested this notion that IFN-γ regulates the PGE2 response using combined transcriptomic and chromatin accessibility analysis. Analysis of our RNAseq data (*Figure 8—figure supplement 1*) revealed two clusters of genes C2 and C5 that were induced by PGE2 but not by IFN-γ (*Figure 8A*); the great majority of these PGE2-inducible genes were inhibited by IFN-γ, and C2 genes were more strongly inhibited than C5 genes. Strikingly, IFN-γ suppressed induction of canonical PGE2-cAMP target genes such as *CREM, HBEGF, HIF1A, PLAUR, KDM6B, AREG, VEGFA*, and genes in RA pathogenic gene modules and expressed in RA synovial macrophage cluster 1 (corresponding to cluster 1 in *Kuo et al., 2019* and *Figure 2*), such as IL-1 and Notch pathway genes and neutrophil chemokines CXCL1/2/3/5/8 (*Figure 8A and B*, *Figure 8—figure supplement 1B*). Thus, IFN-γ broadly suppresses the PGE2 response, including genes important in RA pathogenesis.

We then analyzed our ATACseq data to gain insight into mechanisms underlying negative regulation of PGE2 responses by IFN-γ (*Figure 8—figure supplement 1C*, *Figure 8C*). IFN-γ suppressed 1765 out of 2411 (72%) of PGE2-induced ATACseq peaks (*Figure 8C*, column G2 depicts suppressed peaks). These IFN-γ-suppressed peaks were most significantly enriched in AP-1 and CEBP motifs (*Figure 8D*; the most significantly enriched motif TGAGTCA closely resembles an AP-1 site). ChromVAR analysis confirmed that PGE2 induced AP-1 activity, and this activity was suppressed by IFN-γ (*Figure 8E*). Additionally, NR4A1/2, NFE2L2, and MAF activity was induced by PGE2 and suppressed by IFN-γ. In

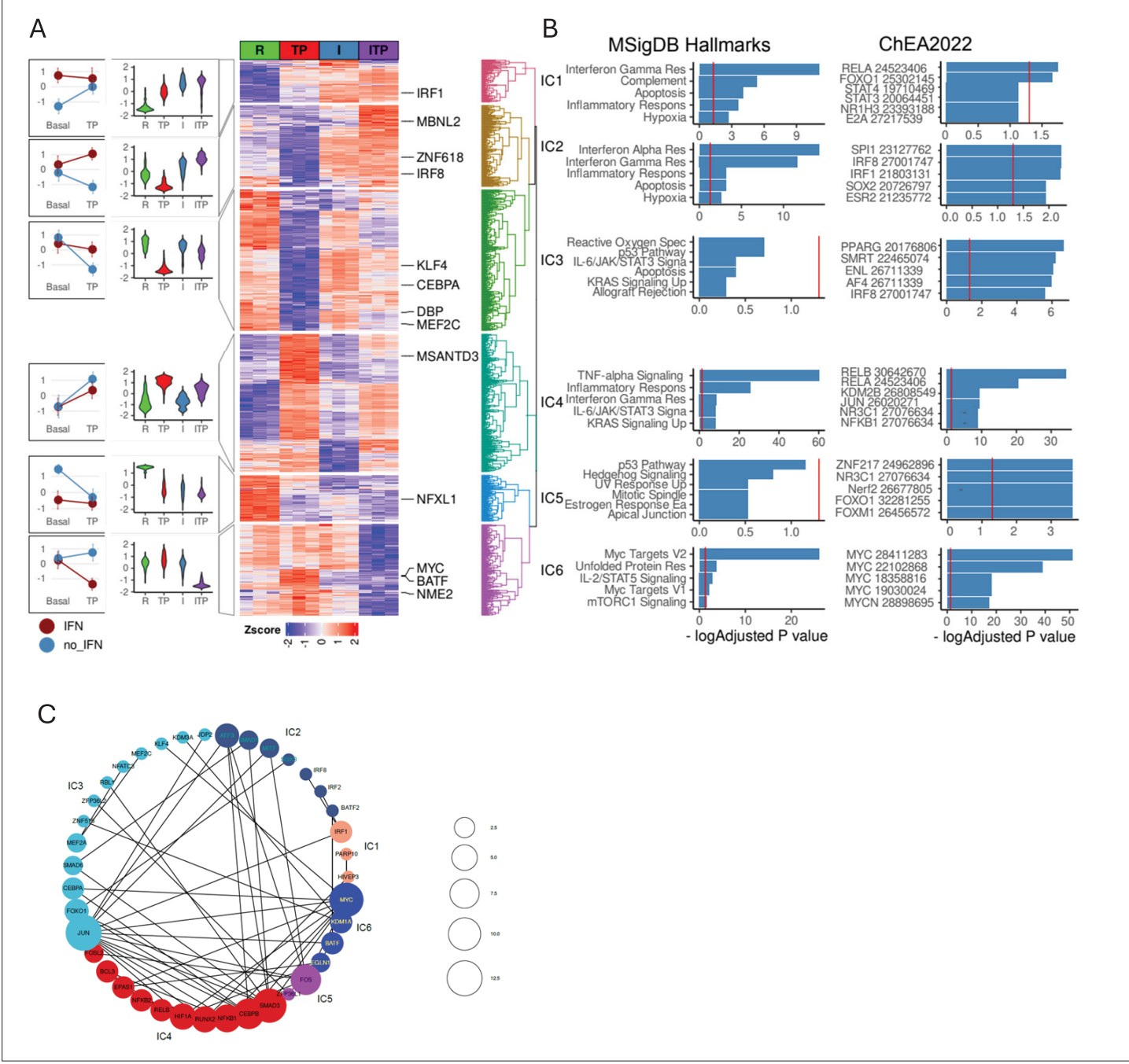

**Figure 7.** Interaction analysis of IFN-γ and the TP response. (**A**) Differentially expressed genes in ((IFN-γ+TP) – IFN-γ) – (IFN-γ – Resting) contrast define statistical interactions between IFN-γ and TP treatments using the RNAseq data, n=3, false discovery rate [FDR] < 0.05, fold change >2. Hierarchical clustering of z-transformed gene expression values (cpm) reveals six interaction clusters (right). Violin plots showing relative gene expression between resting (R), TP, IFN-γ alone (I), and IFN-γ+TP (ITP) conditions (second from left). Interaction plot (left). (**B**) Pathway analysis of the genes in the interaction clusters defined in panel (**A**). (**C**) STRING functional protein association network of transcription factors from each cluster (fold change >4, FDR <0.05). Lines designate functional interactions between individual TFs. The size of nodes is proportional to the number of STRINGDB interactions.

contrast to the HOMER de novo motif analysis described above that focused on IFN-γ-suppressed peaks, ChromVAR analysis, which takes into account all peaks, showed that both PGE2 and IFN-γ induced CEBP activity. Remarkably, at the mRNA expression level, PGE2 and IFN-γ both induced and repressed different members of the CEBP TF family, with coordinate induction of CEBPB, but opposing regulation of CEBPA, CEBPD, and CEBPE (*Figure 8F*). TOBIAS footprinting analysis showed

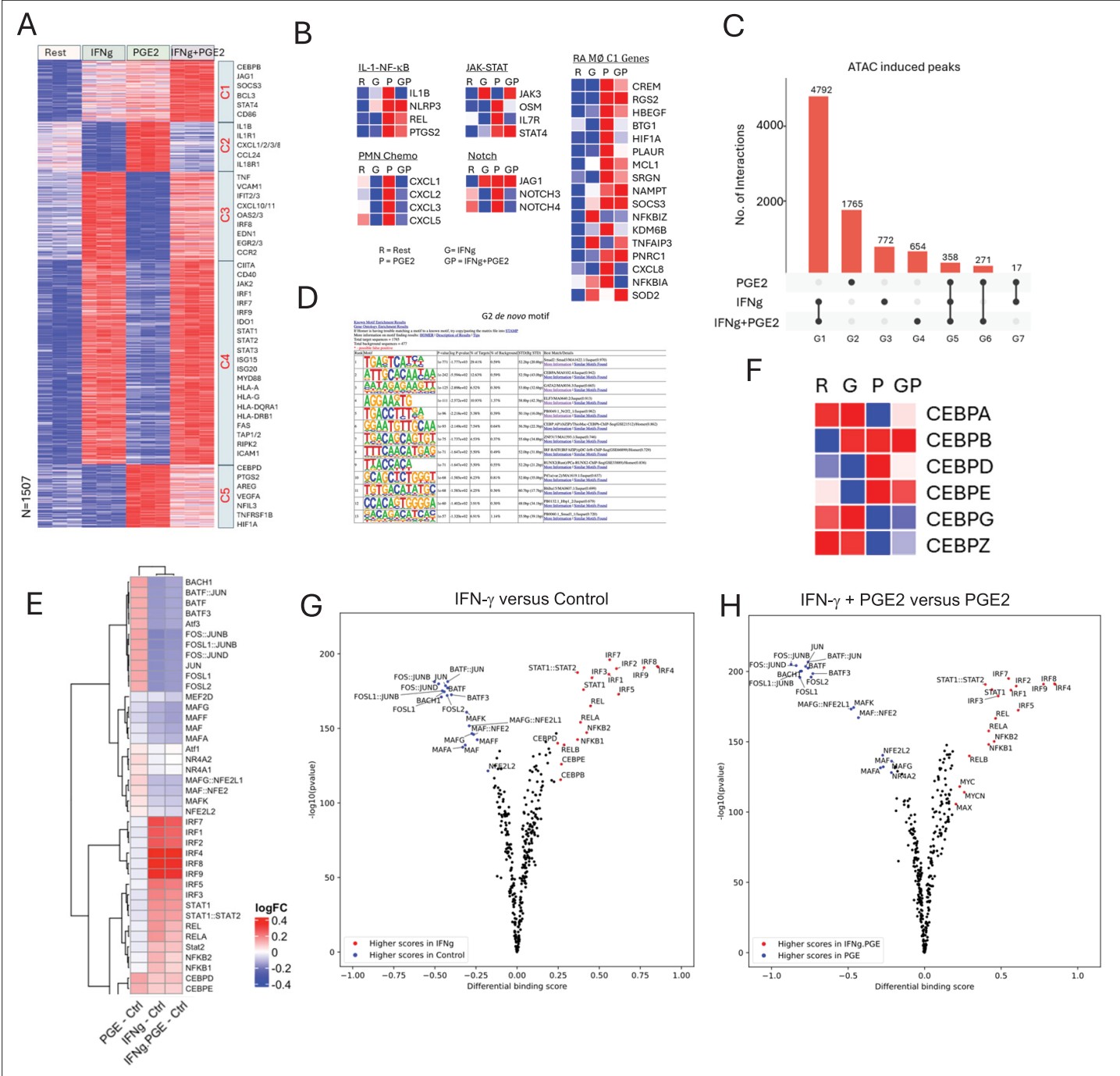

**Figure 8.** IFN-γ inhibits PGE2-induced gene expression and chromatin accessibility. (**A–B, F**) Analysis of RNAseq data obtained using monocytes from three independent donors. (**A**) K-means clustering of differentially upregulated genes in any pairwise comparison relative to resting control (>2-fold induction, false discovery rate [FDR] < 0.05). k=5. (**B**) Heatmaps depicting expression of genes in pathogenic pathways and expressed in C1 RA macrophages as defined in **Figure 2C and D**. (**C–E, G, H**). Analysis of ATACseq data obtained using monocytes from three independent donors. (**C**) UPSET plot of differentially upregulated ATACseq peaks in any pairwise comparison relative to resting control (>2-fold induction, FDR < 0.05). (**D**) De novo motif analysis using HOMER of ATACseq peaks induced uniquely by PGE2 (corresponding to G2 in panel **C**). (**E**) Heatmap of the differential TF activity scores derived from ChromVAR analysis of ATACseq data for P, IFN-γ, or IFN-γ+PGE2 treated monocytes, compared to resting control. (**F**) Heatmap depicting expression of CEBP genes in RNAseq data. (**G, H**) Volcano plots of differential binding analysis of ATACseq peaks of IFN-γ versus resting control (**G**) and IFN-γ+PGE2 versus PGE2 (**H**) conditions using TOBIAS. The IFN-γ versus resting results (**G**) reproduce results in **Mishra and Ivashkiv, 2024** that were obtained in independent experiments with different blood donors.

The online version of this article includes the following figure supplement(s) for figure 8:

**Figure supplement 1.** IFN-γ inhibits PGE2-induced gene expression and chromatin accessibility.

that both PGE2 (see *Figure 5E*) and IFN-γ induced CEBP occupancy (*Figure 8G*); the latter confirms findings in our previous report that used a different independent dataset (*Mishra and Ivashkiv, 2024*). Direct comparison of IFN-γ+PGE2 relative to PGE2 alone using TOBIAS showed striking suppression of AP-1 occupancy and further supported IFN-γ-mediated suppression of NR4A2, NFE2L2, and MAF (*Figure 8H*). Collectively, these results show that IFN-γ inhibits PGE2 responses by suppressing AP-1 and a PGE2-induced transcriptional program mediated by AP-1, NR4A1/2, NFE2L2, and MAF TFs, and suggest that IFN-γ and PGE2 costimulation remodels the composition and genomic binding profile of CEBP proteins to be distinct from that induced by each stimulus alone.

## Discussion

IL-1β-expressing macrophages have been implicated in the pathogenesis of RA, ICI-arthritis, and pancreatic cancer, but the mechanisms that induce these cells and the extent to which they contribute to arthritic phenotypes are not known. In this study, we utilized transcriptomics to identify a 'TNF+PGE2' (TP) signature in ex vivo-stimulated monocytes and showed expression of this signature in previously identified RA macrophage subsets and in ICI-arthritis IL-1$\beta$+ synovial macrophage subsets that were defined by single cell RNA sequencing. We investigated mechanisms of crosstalk between PGE2 and TNF that costimulated expression of inflammatory and pathogenic genes. Epigenomic analysis revealed the cooperation of PGE2-induced AP-1, CEBP, and NR4A family TFs with TNF-induced NF-κB activity to drive expression of pathogenic gene modules including IL-1, Notch, and neutrophil chemokine pathways. The genes in these pathways are distinct from canonical inflammatory genes such as *TNF*, *IL6*, and *IL12B* that are driven by core NF-κB and IRF signaling. Unexpectedly, IFN-γ suppressed AP-1 and NR4A, and altered CEBP activity to ablate induction of IL-1, Notch, and neutrophil chemokine genes, while promoting expression of distinct inflammatory genes such as *TNF* and T cell chemokines such as CXCL10. IFN-γ and PGE2 signaling also appeared to oppose each other in vivo in inflammatory arthritis, based upon mutually exclusive expression of IFN and TP signatures in different cell clusters. These results reveal the basis for synergistic induction of inflammatory genes by PGE2 and TNF, and a novel regulatory axis whereby IFN-γ and PGE2 oppose each other to determine the balance between two distinct TNF-induced inflammatory gene expression programs (*Figure 9A and B*).

TNF and microbial products that engage Toll-like receptors (TLRs) activate a core set of inflammatory genes such as *TNF*, *IL6*, *IFNB*, and *IL12B* that are NF-κB targets and whose expression is amplified by IFNs via induction of IRF factors that cooperate with NF-κB (*Mishra and Ivashkiv, 2024*; *Figure 8A*). Expression of these 'signature' inflammatory genes is also augmented by innate immune training, which is mediated in many contexts by IFNs and IRFs (*Mishra and Ivashkiv, 2024*). In contrast to IFNs that are generally thought to be 'pro-inflammatory' (*Ivashkiv, 2018*), PGE2-cAMP signaling in innate immune cells is considered anti-inflammatory based upon inhibition of *TNF* and ISGs such as *CXCL10* (*Kawahara et al., 2015*; *Tsuge et al., 2019*; *Yokoyama et al., 2013*; *Figure 8A*). Recent reports by our and other laboratories showed that PGE2+TNF drive expression of EGFR ligands such as *HBEGF* in myeloid cells, thereby promoting proliferation and invasive behavior of synovial fibroblasts (*Kuo et al., 2019*), expansion of pancreatic tumors (*Caronni et al., 2023*), and survival and reparative capacities of intestinal epithelial cells (*Zhou et al., 2022*). Thus, TP costimulation can have pathogenic or protective roles, depending on context. The current study further extends this paradigm by showing that TP costimulation promotes expression of additional pathogenic modules in RA including IL-1-related genes that promote tissue degradation, Notch ligands that can activate synovial fibroblasts (*Wei et al., 2020*; *Zack et al., 2024*), and neutrophil chemokines that can contribute to initiation or flares of inflammatory arthritis (*Gravallese and Firestein, 2023*; *Zec et al., 2023*).

Although our study focuses on monocytes/macrophages, TNF and PGE2 also cooperate to promote maturation and antigen-presenting capacity of DCs (*Rieser et al., 1997*). Such mature DCs can contribute to inflammation and autoimmunity (*Steinman et al., 2003*), and it is tempting to speculate that expression of TP genes in RA DC2s promotes their antigen-presenting capability. Additionally, in vitro-matured human DCs, including those stimulated with TNF+PGE2, have been developed for DC-mediated therapy such as vaccination against tumors (*Steinman and Pope, 2002*). In previous work, we had found that in human in vitro generated DCs, TNF and PGE2 costimulation induced expression of DC maturation markers such as CD25, CD40, and CD86 (*Hu et al., 2008*). It will be

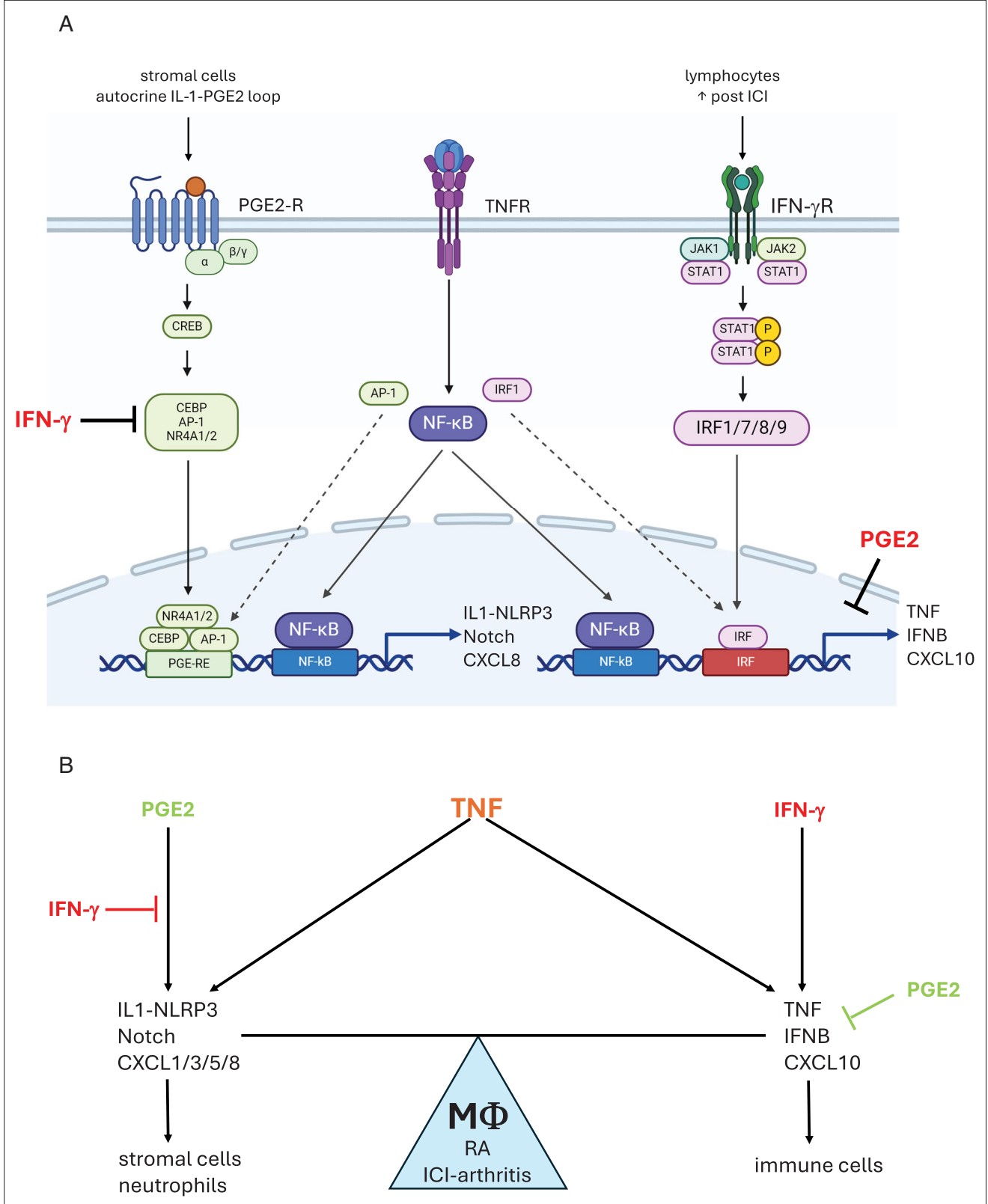

**Figure 9.** Crosstalk between PGE2 and TNF signaling and its regulation by IFN-γ. (**A**) PGE2 costimulates TNF-induced expression of select inflammatory genes such as IL-1 and Notch pathway genes and neutrophil chemokines by inducing transcription factors including CEBP, AP-1, and NR4A1/2 that cooperate with TNF-activated NF-κB (left). IFN-γ induces IRF transcription factors that cooperate with TNF-activated NF-κB to costimulate distinct inflammatory genes such as *TNF* and T cell chemokines such as CXCL10, reviewed in ***Mishra and Ivashkiv, 2024*** (right). IFN-γ inhibits induction of

*Figure 9 continued on next page*

*Figure 9 continued*

TP-costimulated genes by suppressing induction of AP-1 and NR4A1/2, and altering the pattern of expression of CEBP factors. PGE2 suppresses TNF-mediated induction of TNF and ISGs. As PGE2 is produced by stromal cells and IFN-γ is produced by lymphocytes, neighboring cells in inflamed tissues will help determine the macrophage response to TNF. Induction of a subset of CEBP factors by IFN-γ is not depicted. (**B**) IFN-γ and PGE2 oppose each other to regulate the balance between distinct TNF-induced inflammatory responses. PGE2 signaling promotes a response that activates stromal cells via IL-1, EGFR ligands, and Notch pathways, and promotes recruitment of neutrophils. IFN-γ suppresses these pathways and instead promotes inflammation via TNF and recruitment of T cells.

interesting in future work to determine whether some of the crosstalk mechanisms described in the current study also play a role in driving gene expression during DC maturation.

In contrast to the cooperation of IRFs with NF-κB that drives signature inflammatory genes, our results suggest that the distinct TP gene expression program is promoted by cooperation of NF-κB with CEBP, AP-1, and NR4A1/2 TFs (*Figure 9A*, left). CEBP is a pioneer TF that can open chromatin, and its induction helps explain the formation of de novo enhancers under costimulation conditions. The above-mentioned PGE2-induced TFs are well-established target genes for canonical cAMP-PKA-CREB signaling (*Altarejos and Montminy, 2011*; *Gerlo et al., 2011*; *Zhang et al., 2005*), which is in accord with our results implicating this pathway downstream of PGE2 and EP2 and EP4 receptors. A role for CREB was not identified by our experimental approaches, possibly because the number of direct CREB target genes was too small to pass statistical significance thresholds or because CREB binds to its target genes under basal conditions and is activated by phosphorylation rather than changes in DNA-binding (*Altarejos and Montminy, 2011*). Interestingly, IFN-γ suppressed the PGE2-induced transcriptional program by inhibiting the activity of AP-1 and NR4A1/2 and altering the expression profile of CEBP, which was associated with abrogation of expression of IL-1, Notch, and neutrophil chemokine gene modules. These results identify an unexpected dual role for IFN-γ and suggest that the inflammatory profile of TNF signaling is regulated by the local balance between PGE2 and IFN-γ activity (*Figure 9B*).

Previous work characterizing RA synovial cells using scRNAseq identified various inflammatory monocyte and macrophage subsets, and reparative MERTK+ macrophages that are associated with remission (*Alivernini et al., 2020*; *Kuo et al., 2019*; *Zhang et al., 2023*; *Zhang et al., 2019*). MERTK+ cells appear to have a tissue-resident macrophage phenotype, produce inflammation-resolving mediators, and promote a repair response in synovial fibroblasts (*Alivernini et al., 2020*). Early studies (*Kuo et al., 2019*; *Zhang et al., 2019*) with limited cell numbers resolved RA synovial myeloid cells into four subtypes, which included inflammatory IL1β+ monocytes that express NF-κB target genes and a distinct monocyte cluster that expresses high levels of ISGs and thus an IFN signature. The inflammatory IL1β+ monocytes also expressed TNF-inducible genes whose expression is augmented by PGE2-producing fibroblasts. In the current study, we have defined TP genes whose expression is co-stimulated by adding PGE2 to TNF and showed that they are expressed in IL1β+ cluster 1 as defined in *Kuo et al., 2019*; *Figure 2A–D*, and in a subset of ICI-arthritis myeloid cells (*Figure 2E and F*). We have also found that the TP signature is expressed in an IL1β+ FCN1+ HBEGF+ subset of RA monocytes defined in a more recent study with larger cell numbers that subclustered RA myeloid cells at a higher resolution of 15 clusters (*Zhang et al., 2023*). However, the TP signature was not limited to IL1β+ FCN1+ HBEGF+ cells, but instead partially extended into adjacent SPP1+, MERTK+ LIVE1-, and DC2 clusters (*Figure 3*); DC2s have recently been shown to have distinct homeostatic and pathogenic functions in RA (*MacDonald et al., 2024*). One possibility is that the proximity of cells to PGE2-producing fibroblasts modulates subsets of functionally distinct SPP1+ and MERTK+ LIVE1- cells to exhibit a superimposed TP phenotype. Alternatively, the TP signature may identify one functional phenotype of synovial monocyte that was not well captured by the clustering as performed in *Zhang et al., 2023*. The latter possibility is suggested by the clean separation of a TP-expressing cluster in ICI-arthritis myeloid cells (*Figure 2*), and the localization of TP-expressing ICI-arthritis cells in one UMAP region that spans several RA clusters (*Figure 3D*). Additionally, the pattern of TP gene expression on these UMAPs very closely resembles the pattern of myeloid cell neighborhoods associated with the myeloid subtype of RA, as shown in Figure 4B of *Zhang et al., 2023*. The function of TP signature-expressing cells will be investigated in future work using functional analysis of cell subtypes isolated from inflamed joints.

Our in vitro studies showing opposition between IFN-γ and PGE2 in regulation of specific TNF-inducible gene sets, suppression of ISGs by PGE2, and suppression of PGE2 responses by IFN-γ predicts that IFN-γ and PGE2 may cross-inhibit each other in vivo, resulting in cell clusters that are ISG+ TP- (strongly activated by IFNs) or ISG- TP+ (strongly activated by PGE2). This prediction was indeed borne out, as the IL1$\beta$+ FCN1+ HBEGF+ cells are essentially devoid of ISG expression (*Zhang et al., 2023*; https://immunogenomics.io), and STAT1+ CXCL10+ ISG-hi cells did not express TP genes (*Figure 3*). As synovial IFN-γ is made predominantly by T cells and PGE2 by fibroblasts, this suggests that colocalization of monocytes with either T cells (e.g., in lymphoid aggregates) or with fibroblasts (e.g., in synovial lining) can determine IFN versus TP phenotype. Cells that simultaneously receive both IFN and PGE2 inputs may exhibit mixed expression of both signatures, for example the TP+ subset of SPP1+ or MERTK+ LYVE1- cells. The relative strength of IFN-γ versus PGE2 signaling would determine functional outcomes. This idea can be tested in future work using spatial transcriptomics to colocalize various myeloid subsets with IFN-γ-expressing T cells or PGE2-producing fibroblasts.

Our results indicate a level of similarity between ICI-arthritis and RA synovial myeloid subsets (*Figure 3B*; similar results were obtained using an independent computational approach to integrate the RA and ICI-arthritis datasets). The striking segregation of monocytic clusters expressing TP signature genes and ISGs in our ICI-arthritis dataset represents another similarity to RA. These results are subject to several caveats. RA myeloid cells were derived from synovial tissue, whereas the majority of ICI-arthritis cells were from synovial fluids. This may explain the lack of more mature tissue macrophage-like cells (clusters M0-M3 in *Figure 3A and B*) in the ICI-arthritis dataset. Additionally, the smaller number of ICI-arthritis cells analyzed limited the ability to capture more rare subsets, such as some of the DC populations that were detected in RA. Lastly, reference mapping, while computationally rigorous, is a constrained approach because it will only assign cells to clusters in the reference set (i.e., will not generate de novo clusters), creating the possibility of false-positive assignment. We decreased this possibility by using only myeloid cells from each dataset and corroborated the results using a complementary dataset integration approach. In future work, it will be important to perform a more thorough analysis of more extensive RA and ICI-arthritis datasets. It will be informative to perform this analysis in responders and nonresponders to TNF blockade and NSAID therapy, which would causally link these pathways with myeloid subsets and help in utilizing myeloid pathogenic cell clusters in patient stratification.

Major sources of PGE2 in inflammatory arthritis and tumors are stromal cells such as fibroblasts and tumor cells (*Bayerl et al., 2023*; *Gong et al., 2023*; *Kawahara et al., 2015*; *Kuo et al., 2019*; *Tsuge et al., 2019*). This argues that macrophages in tissue niches characterized by interaction with fibroblasts will strongly express the PGE2-driven gene expression program including expression of EGFR and Notch ligands that activate fibroblasts, suggestive of a positive feedback loop. Associated high expression of TP-induced neutrophil chemokines would promote neutrophil-mediated inflammation. IL-1 action on myeloid cells strongly induces COX2 and thus PG production, and an autocrine IL-1-PGE2 loop can further amplify the TP response, as shown in pancreatic tumors (*Caronni et al., 2023*). In contrast, the major sources of IFN-γ in RA are T cells, which are localized either to ectopic germinal center-like structures or are dispersed throughout inflamed synovium (*Gravallese and Firestein, 2023*). IFN-γ is also highly expressed by CD8+T cells in ICI-arthritis (*Kim et al., 2022*; *Wang et al., 2023*). IFN-γ-exposed macrophages express T cell chemokines such as CXCL10 instead of neutrophil chemokines, thereby promoting T cell-mediated pathogenesis instead of stromal-neutrophil mediated pathogenic responses. Based upon our previous work (*Mishra and Ivashkiv, 2024*; *Qiao et al., 2013*) and the current study, these macrophages would be predicted to express signature NF-κB target genes such as *TNF*, which was indeed the case with the STAT1+ CXCL10+ RA cluster. Distinct subsets of RA patients may exhibit fibroblast/PGE2 versus T cell/IFN-γ-dominant phenotypes and thus be differentially responsive to therapeutics.

In RA, macrophages are major producers of TNF, which is a well-established therapeutic target. However, only a subset of RA patients show >50% improvement after TNF blockade therapy, and even these patients can become resistant to therapy (*Perera et al., 2024*). One current explanation for resistance to TNF blockade therapy is that in subsets of RA patients synovitis is driven by distinct pathogenic mechanisms; additionally, these different mechanisms can emerge when TNF activity is blocked, which can explain acquired resistance to TNF blockade. Such distinct mechanisms of RA pathogenesis include those driven by IL-6 and IL-1 instead of TNF (*Gravallese and Firestein, 2023*;

*McInnes and Schett, 2011*). While IL-6 blockade therapy is highly effective in a subset of RA patients, IL-1 blockade with anakinra, although an FDA-approved therapy for RA that is effective at suppressing tissue degradation, appears to have limited efficacy in suppressing inflammatory symptoms. One explanation for potential low efficacy in suppressing symptoms of inflammation such as joint swelling is that IL-1 pathways are not primarily coupled with symptoms of inflammation, but instead with tissue degradation via their effects on synovial fibroblasts. Alternatively, our work raises the intriguing possibility that interrupting the IL-1-PGE2 axis using IL-1 blockade will release genes such as *TNF* and *CXCL10* from suppression and result in the emergence of stronger TNF- and lymphocyte-driven pathogenic pathways that maintain synovitis. If substantiated, this notion would promote the idea of combination therapy of using anakinra with either TNF blockade or T cell-targeted therapy.

# Materials and methods

## Primary human monocyte isolation and culture

Deidentified buffy coats were purchased from the New York Blood Center following a protocol (# 2016-958) approved by the Hospital for Special Surgery Institutional Review Board. Peripheral blood mononuclear cells (PBMCs) were isolated using density gradient centrifugation with Lymphoprep (Accurate Chemical), and monocytes were purified with anti-CD14 magnetic beads from PBMCs immediately after isolation as recommended by the manufacturer (Miltenyi Biotec). Monocytes were cultured at 37°C, 5% $CO_2$ in RPMI-1640 medium (Invitrogen) supplemented with 10% heat-inactivated defined fetal bovine serum (FBS) (HyClone Fisher), penicillin-streptomycin (Invitrogen), L-glutamine (Invitrogen), and 20 ng/ml human macrophage-colony stimulating factor (M-CSF).

## Mouse bone marrow-derived macrophage culture

Animal experiments were approved by the Weill Cornell Medicine IACUC Committee (protocol # 2015-0055). Male C57BL/6J mice at 6–8 weeks old were purchased from the Jackson Laboratories and housed under specific pathogen-free conditions. Bone marrow cells were harvested after euthanasia by $CO_2$ asphyxiation and cultured in RPMI-1640 medium (Invitrogen) supplemented with 10% heat-inactivated defined FBS (HyClone Fisher), penicillin-streptomycin (Invitrogen), L-glutamine (Invitrogen), and 20 ng/ml mouse M-CSF.

## Analysis of mRNA amounts (qPCR)

Total RNA was isolated using the RNeasy Mini Kit (QIAGEN) following the manufacturer's instructions. Reverse transcription of RNA into complementary DNA (cDNA) was performed using the RevertAid RT Reverse Transcription Kit (Thermo Fisher Scientific) according to the manufacturer's protocol, and the resulting cDNA was used for downstream analysis. For quantitative real-time PCR (qPCR), Fast SYBR Green Master Mix (Applied Biosystems) and a QuantStudio5 Real-time PCR system (Applied Biosystems) were used. CT values obtained from qPCR were normalized to the housekeeping gene *GAPDH*. The primers used for qPCR were: CXCL10: F: 5'-ATTTGCTGCCTTATCTTTCTG-3' R: 5'-TCTC ACCCTTCTTTTTCATTGTAG-3'; TNF F: 5'-AATAGGCTGTTCCCATGTAGC-3' R: 5'-AGAGGCTCAGCA ATGAGTGA-3'; GAPDH: F: 5'- ATCAAGAAGGTGGTGAAGCA-3' R: 5'-GTCGCTGTTGAAGTCAGAGG A-3'; IL1B: F: 5'- TTCGACACATGGGATAACGAGG; R: 5'-TTTTTGCTGTGAGTCCCGGAG; CSF3: F: 5'-CCAGGAGAAGCTGGTGAGTG; R: 5'-GAAAAGGCCGCTATGGAGTT; HBEGF: 5'- F:AGGAGCAC GGGAAAAGAAAG; R: 5'-CTCAGCCCATGACACCTCTC; IL1A: F: 5'-AGTAGCAACCAACGGGAAGG ; R: 5'-AAGGTGCTGACCTAGGCTTG; STAT4: F: 5'-GAGACCAGCTCATTGCCTGT; R: 5'-CAATGTGG CAGGTGGAGGAT. Relative expression of target genes was calculated using the ΔCt method, where ΔCt represents the difference in threshold cycle values between the target gene and *GAPDH*. The results are presented as a percentage of *GAPDH* expression ($100/2^{\Delta Ct}$). Data points were not omitted from analysis.

## Western blotting

Primary human monocytes were washed with cold PBS after indicated treatments and harvested in 50 ul cold lysis buffer containing Tris-HCl pH 7.4, NaCl, EDTA, Triton X-100, $Na_3VO_4$, phosSTOP EASYPACK, Pefabloc, and EDTA-free complete protease inhibitor cocktail. After a 10 min ice incubation, cell debris was pelleted at 16,000×*g* at 4°C for 10 min. The soluble protein fraction was

combined with 4×Laemmli sample buffer containing 2-mercaptoethanol and subjected to SDS-PAGE (electrophoresis) on 4–12% Bis-Tris gels. Following the transfer of gels to polyvinylidene difluoride membranes, membranes were blocked in 5% (w/v) bovine serum albumin in TBS with Tween-20 (TBST) at room temperature for at least 1 hr. Incubation with primary antibodies (diluted 1:1000 in blocking buffer) was performed overnight at 4°C. The p38 antibodies were from Cell Signaling Technology (cat# 9212) and Stat4 antibodies (C46B10 rabbit mAb) were from Cell Signaling Technology (cat# 2653). Membranes were washed three times with TBST and probed with anti-rabbit IgG secondary antibodies conjugated to horseradish peroxidase (diluted 1:5000 in blocking buffer) for 1 hr at room temperature. Enhanced chemiluminescent substrates (ECL western blotting reagents (PerkinElmer, cat# NEL105001EA)) were used for detection, followed by visualization on autoradiography film (Thomas Scientific, cat# E3018). Restore PLUS western blotting stripping buffer (Thermo Fisher Scientific) was applied for membranes requiring sequential probing with different primary antibodies.

## RNA sequencing and data analysis

Libraries for sequencing were prepared using mRNA that was enriched from total RNA using NEBNext Poly(A) mRNA Magnetic Isolation Module, and enriched mRNA was used as an input for the NEBNext Ultra II RNA Library Prep Kit (New England Biolabs [NEB]), following the manufacturer's instructions. Quality of all RNA and library preparations was evaluated with BioAnalyser 2100 (Agilent). Libraries were sequenced by the Genomic Resources Core Facility at Weill Cornell Medicine, obtaining 50 bp single-end or paired-end reads to a depth of ~15–20 million reads per sample. Read quality was assessed and adapters trimmed using fastp. Reads were then mapped to the human genome (hg38) using STAR aligner and reads in exons were counted against Gencode v37 with Featurecount. Differential gene expression analysis was performed in R with edgeR using the quasi-likelihood framework. Only genes with expression levels exceeding 4 counts per million reads in at least one group were used for downstream analysis. Benjamini–Hochberg FDR procedure was used to correct for multiple testing. Genes were categorized as upregulated if log2FC ≥1 and FDR ≤ 0.05 threshold was satisfied, downregulated if log2FC ≤ −1 and FDR ≤ 0.05. Heatmaps were obtained using the Morpheus web application and replotted using the R package pheatmap. To analyze the regulatory factors involved in IFN-γ - (T+P) interactions, we selected all TFs based on *Lambert et al., 2018* from the interacting clusters (*Figure 6*) and used them to build a protein associating network with StringDB (*Szklarczyk et al., 2019*) limiting the associations to the highest interaction score (0.9) and visualized with R *tidygraph* framework using circular layout. RNAseq experiments were performed using different blood donors: (1) monocytes stimulated by vehicle control or T, P, or TP for 3 or 24 hr (three donors). (2) Monocytes primed with or without IFN-γ and then stimulated with vehicle control, T, P, or TP for 3 hr (three different donors). Aliquots of cells from this experiment were used for ATACseq.

## ATAC sequencing and data analysis

One million monocytes were lysed using cold lysis buffer (10 mM Tris-HCl, pH 7.4, 10 mM NaCl, 3 mM MgCl$_2$, and 0.1% IGEPAL CA-630), and nuclei were immediately pelleted at 500×$g$ for 10 min in a refrigerated centrifuge. The pellet was resuspended in a transposase reaction mix consisting of 25 μl 2×TD buffer, 2.5 μl transposase (Illumina), and 22.5 μl nuclease-free water. The transposition reaction was carried out for 30 min at 37°C. Following transposition, the sample was purified using a MinElute PCR Purification kit. Library fragments were amplified using 1×NEB next PCR master mix and 1.25 M custom Nextera PCR primers, with subsequent purification using a QIAGEN PCR cleanup kit, yielding a final library concentration of ~30 nM in 20 μl. Libraries were amplified for a total of 10–13 cycles and subjected to high-throughput sequencing at the Genomic Resources Core Facility at Weill Cornell Medicine with 50 bp paired-end reads. Data on ATAC-seq experiments were derived from three independent experiments with different blood donors.

For ATACseq data analysis, read alignments were performed against the GRCh38/hg38 reference human genome. Peak calling was conducted using MACS2 with the following parameters: "macs2 callpeak -f BAMPE -g hs -q 0.01 `--nomodel --shift 37 --extsize` 76 ". A master consensus peak set was generated by merging the resulting peak files for each treatment condition, followed by merging peaks within 50 bp of each other. Quantification of peaks to compare global ATACseq signal changes in the BAM files was conducted using the NCBI/BAMscale program. Raw count matrices were obtained utilizing the BAMscale program. Subsequent analysis utilized the HSS Genomic Center

Bioinformatic Core's ATACseq analysis pipeline (https://gitlab.com/hssgenomics/Shiny-ATAC; *Oliver, 2023*) for peak filtering, annotation relative to genomic features, differential peak analysis, and enrichment of signal around specific motifs using ChromVAR (*Schep et al., 2017*). Footprint analyses were performed using TOBIAS (*Bentsen et al., 2020*) according to the user manual. Briefly, we filtered the JASPAR2022-CORE_vertebrates TF list based on the TF expressed in our RNAseq data and used this as TF input for motif footprinting. De novo TF motif enrichment analysis was carried out using the motif finder program findMotifsGenome in the HOMER package, focusing on the given peaks. Peak sequences were compared to random genomic fragments of the same size and normalized G+C content to identify enriched motifs in the targeted sequences.

## ICI-arthritis sample preparation for single-cell RNA sequencing

All research using patient samples at the Hospital for Special Surgery adhered to approved protocols (IRB protocols 2017-1898 and 2017-1871) with informed consent as required; samples investigated in this study correspond to a subset of those subjected to a distinct analysis in *Wang et al., 2023*. Synovial fluids were collected from ICI-arthritis as discarded fluid during patient visits. Tissue samples were collected during arthroplasty surgery. Mononuclear cells were isolated from synovial fluid by density centrifugation using Ficoll-Paque Plus (GE Healthcare), preserved in Cryostor CS10 (Stemcell Technologies), and placed in liquid nitrogen for long-term storage. Tissues received post-surgery were processed into fragments of 2–3 mm$^3$, preserved in Cryostor CS10, and placed in liquid nitrogen for long-term storage.

For sample preparation for single-cell RNA sequencing, tissues were thawed into pre-warmed media at 37°C. The media composition was RPMI 1640 (Corning) containing 10% defined FBS (Cytiva) and 1% of 200 mM L-glutamine (Gibco). The tissues were then washed twice in RPMI alone for 5 min and chopped into fine pieces using a blade. Finely cut tissue was subjected to 30 min in a digestion buffer composed of RPMI, Liberase TL (100 µg/ml; Roche) and DNaseI (100 µg/ml; Roche). The fragments were digested in 5 ml polystyrene tubes (5 ml/sample of digestion buffer), securely placed in a MACSmix tube rotator (Miltenyi Biotec) in an incubator at 37°C with 5% $CO_2$. The tissue preparation was filtered through a 70 µM cell strainer (BD) and mashed using the back of a syringe plunger. The eluate containing the cells was washed with media and centrifuged at 1500 rpm for 4 min at 4°C. The pellet was resuspended in media and filtered again through a 40 µM cell strainer (BD). Cells in the eluate were then counted and used in downstream experiments. Fluid samples were thawed from cold storage into pre-warmed media at 37°C and then processed similarly to tissue samples. Live mononuclear cells were then FACS-sorted from five synovial fluid and two synovial tissue samples. Cells were sorted on a three-laser BD FACS Aria Fusion cell sorter at the Flow Cytometry Core Facility at Weill Cornell Medicine (WCM). Intact cells were gated according to forward scatter and side scatter area (FSC-A and SSC-A). Doublets were excluded by serial FSC-H/FSC-W and SSC-H/SSC-W gates (H, height; W, width). Non-viable cells were excluded based on DAPI uptake, sorted through a 70 µM nozzle at 70 523 psi. Flow cytometric quantification of cell populations was performed using FlowJo v.10.0.7.

## ICI-arthritis single-cell RNA sequencing and data analysis

3′ gene expression (GEX) libraries of live synovial mononuclear cells were prepared using Chromium Single Cell 3′ v3 Kit reagents and protocols provided by 10X Genomics. The pooled libraries at 10 nM concentration were sequenced using NovaSeq6000 S2 Flow Cell using the Illumina platform at Genomics Research Core Facility at WCM. 10× FASTQ files were processed with the Cellranger count 4.0 pipeline with default parameters. Reads were aligned to the human reference sequence GRCh38. Seurat package (v.4.0.0) was used to perform unbiased clustering. The 3′ GEX dataset of sorted live synovial cells had QC performed to also remove cells with less than 150 genes, more than 7500 genes, or >25% mitochondrial gene expression, resulting in a total of 44,959 cells and 26,424 genes. This dataset was then log-normalized using a scale factor of 10,000. Potential confounders such as percent mitochondrial gene expression and number of UMI per cell were regressed out during scaling (mean of 0 and variance of 1) for 3′ GEX dataset.

Principal component analysis was used with the top 2000 highly variable genes. Elbow plot was used to determine the statistically significant principal components of 17 PCs for follow-up analysis. Harmony (v1.0) was performed to improve integration and correct for batch effects on

our samples, with parameters of max.iter.cluster=30, and max.iter.harmony=20 and sample as the only covariate. Eleven clusters for the 3′ dataset at 0.2 resolution were found, and their identity was annotated based on the expression of differentially expressed genes (DEGs) using the FindAllMarkers function using default parameters. Additional grouping of the 3′ GEX dataset of select clusters at resolution 0.2 was used to define only monocytes and macrophages to create a new data object. These clusters contained 14,110 cells after going through linear dimensional reduction of the top 5 PCs with Harmony as described above. Eight clusters were defined at 0.3 resolution for the final analysis.

To compare the ICI-arthritis scRNAseq data to a previously reported RA CITEseq dataset, we first retrieved the RA myeloid cell data from *Zhang et al., 2023*; available on Synapse syn52297840. Reference mapping was performed using the uwot model (https://jlmelville.github.io/uwot/authors.html) provided in syn52297840 with Seurat's MapQuery function as described (*Hao et al., 2024*). For analysis of TP gene expression in RA and ICI-arthritis myeloid cells, gene set activity scores were calculated using the Bioconductor tool AUCell (v1.24.0) in R (v4.3.2) on the scRNA-seq datasets. AUCell ranks genes based on their expression in each cell and computes the area under the curve (AUC) to quantify the enrichment of predefined gene sets. The threshold for determining active gene sets (corresponding to the pink and red dots in *Figure 3*) was automatically set by the AUCell algorithm using the default settings (AUCell_buildRankings and AUCell_calcAUC functions), which identifies the ranking threshold based on the distribution of expression ranks across cells (*Aibar et al., 2017*).

## Statistical analysis

GraphPad Prism was used for all statistical analysis. Information about the specific tests used and number of independent experiments is provided in the figure legends.

## Data and code availability

Sequencing data from this study have been deposited at GEO (RNAseq: GSE272019; ATACseq: GSE272017) and will be publicly available from the date of publication. The scRNAseq data object is available at https://gitlab.com/hssgenomics/icia_myeloid.

## Acknowledgements

We thank the Weill Cornell Medicine Genomics Core for sequencing and the Computational Biology Core of the David Z Rosensweig Genomics Center at HSS for data analysis. This work was supported by grants AR046713, AI046712, and AR050401 from the NIH (to LBI). The Rosensweig Genomics Center is supported by the Tow Foundation.

## Additional information

### Funding

| Funder | Grant reference number | Author |
|---|---|---|
| National Institute of Arthritis and Musculoskeletal and Skin Diseases | AR046713 | Lionel B Ivashkiv |
| National Institute of Arthritis and Musculoskeletal and Skin Diseases | AR050401 | Lionel B Ivashkiv |
| National Institute of Allergy and Infectious Diseases | AI046712 | Lionel B Ivashkiv |

The funders had no role in study design, data collection and interpretation, or the decision to submit the work for publication.

## Author contributions

Upneet K Sokhi, Data curation, Investigation, Methodology, Writing – review and editing; Ruoxi Yuan, Bikash Mishra, Formal analysis, Visualization, Writing – review and editing; Yurii Chinenov, Formal analysis, Visualization, Methodology, Writing – original draft, Writing – review and editing; Anvita Singaraju, Formal analysis, Investigation, Visualization, Writing – original draft, Writing – review and editing; Karmela K Chan, Anne Bass, Investigation, Writing – review and editing; Richard D Bell, Formal analysis, Writing – review and editing; Laura Donlin, Conceptualization, Supervision, Investigation, Writing – review and editing; Lionel B Ivashkiv, Conceptualization, Supervision, Funding acquisition, Investigation, Writing – original draft, Writing – review and editing

## Author ORCIDs

Upneet K Sokhi ⓘ https://orcid.org/0000-0001-5907-5646
Lionel B Ivashkiv ⓘ https://orcid.org/0000-0002-9951-0646

## Ethics

All research using patient samples was approved by the Hospital for Special Surgery IRB (protocols # 2016-958, 2017-1898, 2017-1871) and informed consent was obtained.
Animal experiments were approved by the Weill Cornell Medicine IACUC Committee (protocol # 2015-0055) and performed in accordance with recommendations in the Guide for the Care and Use of Laboratory Animals of the National Institutes of Health.

Reviewer #1 (Public review): https://doi.org/10.7554/eLife.104367.3.sa1
Reviewer #2 (Public review): https://doi.org/10.7554/eLife.104367.3.sa2
Author response https://doi.org/10.7554/eLife.104367.3.sa3

---

# Additional files

## Supplementary files

Supplementary file 1. Table listing the clinical characteristics of ICI-arthritis patients whose samples were used in the scRNAseq experiments.

MDAR checklist

## Data availability

Sequencing data have been deposited in GEO under accession codes GSE272019 and GSE272017. The scRNAseq data object is available at https://gitlab.com/hssgenomics/icia_myeloid (copy archived at *Yuan, 2025*).

The following datasets were generated:

| Author(s) | Year | Dataset title | Dataset URL | Database and Identifier |
|---|---|---|---|---|
| Sokhi UK, Yuan R, Mishra B, Chinenov Y, Singaraju A, Chan KK, Bass AR, Bel RD, Donlin L, Ivashkiv LB | 2025 | Opposing Regulation of TNF Responses and IL-1b+ Macrophages by PGE2-cAMP Signaling and IFN-g [RNA-seq] | http://www.ncbi.nlm.nih.gov/geo/query/acc.cgi?acc=GSE272019 | NCBI Gene Expression Omnibus, GSE272019 |
| Sokhi UK, Yuan R, Mishra B, Chinenov Y, Singaraju A, Chan KK, Bass AR, Bel RD, Donlin L, Ivashkiv LB | 2025 | Opposing Regulation of TNF Responses and IL-1b+ Macrophages by PGE2-cAMP Signaling and IFN-g [ATAC-seq] | http://www.ncbi.nlm.nih.gov/geo/query/acc.cgi?acc=GSE272017 | NCBI Gene Expression Omnibus, GSE272017 |

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
