## [Editor Report · eLife Assessment]

The article contains **important** findings regarding inflammatory macrophage subsets that have theoretical and/or practical applications beyond the field of rheumatology. The authors demonstrate with **compelling** evidence the effects of PGE2 on TNF signaling. This work will be of broad interest to immunologists and cell biologists.

---

## [Referee Report · Reviewer #1 (Public review)]

Summary:

This article investigates the phenotype of macrophages with a pathogenic role in arthritis, particularly focusing on arthritis induced by immune checkpoint inhibitor (ICI) therapy.

Building on prior data from monocyte-macrophage coculture with fibroblasts, the authors hypothesized a unique role for the combined actions of prostaglandin PGE2 and TNF. The authors studied this combined state using an in vitro model with macrophages derived from monocytes of healthy donors. They complemented this with single-cell transcriptomic and epigenetic data from patients with ICI-RA, specifically, macrophages sorted out of synovial fluid and tissue samples. The study addressed critical questions regarding the regulation of PGE2 and TNF: Are their actions co-regulated or antagonistic? How do they interact with IFN-γ in shaping macrophage responses?

This study is the first to specifically investigate a macrophage subset responsive to the PGE2 and TNF combination in the context of ICI-RA, describes a new and easily reproducible in vitro model, and studies the role of IFNgamma regulation of this particular Mф subset.

Strengths:

Methodological quality: The authors employed a robust combination of approaches, including validation of bulk RNA-seq findings through complementary methods. The methods description is excellent and allows for reproducible research. Importantly, the authors compared their in vitro model with ex vivo single-cell data, demonstrating that their model accurately reflects the molecular mechanisms driving the pathogenicity of this macrophage subset.

Comments on latest version:

The revisions made to this manuscript followed the suggestions and improved the manuscript. The authors have thoroughly addressed my previous concerns, making several key improvements:

The expanded comparison between rheumatoid arthritis (RA) and immune checkpoint inhibitor-induced RA (ICI-RA) in both cellular and molecular pathology is excellent. These additions to the literature review and discussion sections significantly strengthen the manuscript and provide valuable context.

I particularly appreciate the added effort in mapping a particular cell subset onto previously published single-cell RNA-Seq embeddings. The enhanced UMAPs with cell subset projection analyses are methodologically compelling, informative and visually are easy to understand for any reader. The new Figure 3 represents a substantial improvement.

More detailed comparisons with previously published single-cell datasets from 2019, 2020, and 2023 effectively contextualize this research within the broader field of rheumatoid arthritis pathogenesis. This enhances the manuscript's value for specialists in autoimmunity and myeloid immunology.

I find the authors' suggestion to use the defined myeloid pathogenic phenotypes as biomarkers for therapy response prediction or dose optimization particularly insightful and clinically relevant.

Overall, the authors have significantly improved both the analysis and presentation of results. The manuscript has been substantially enhanced.

---

## [Referee Report · Reviewer #2 (Public review)]

Summary/Significance of the findings:

The authors have done a great job by extensively carrying out transcriptomic and epigenomic analyses in the primary human/mouse monocytes/macrophages to investigate TNF-PGE2 (TP) crosstalk and their regulation by IFN-γ in the Rheumatoid arthritis (RA) synovial macrophages. They proposed that TP induces inflammatory genes via a novel regulatory axis whereby IFN-γ and PGE2 oppose each other to determine the balance between two distinct TNF-induced inflammatory gene expression programs relevant to RA and ICI-arthritis.

Strengths:

The authors have done a great job on RT-qPCR analysis of gene expression in primary human monocytes stimulated with TNF and showing the selective agonists of PGE2 receptors EP2 and EP4 22 that signal predominantly via cAMP. They have beautifully shown IFN-γ opposes the effects of PGE2 on TNF-induced gene expression. They found that TP signature genes are activated by cooperation of PGE2-induced AP-1, CEBP, and NR4A with TNF-induced NF-κB activity. On the other hand, they found that IFN-γ suppressed induction of AP-1, CEBP, and NR4A activity to ablate induction of IL-1, Notch, and neutrophil chemokine genes but promoted expression of distinct inflammatory genes such as TNF and T cell chemokines like CXCL10 indicating that TP induces inflammatory genes via IFN-γ in the RA and ICI-arthritis.

Comments on latest version:

The authors have answered my questions and i recommend this manuscript for publication.

---

## [Author Response]

The following is the authors’ response to the original reviews

**Public Reviews:**

**Reviewer #1 (Public review):**
Summary:This article investigates the phenotype of macrophages with a pathogenic role in arthritis, particularly focusing on arthritis induced by immune checkpoint inhibitor (ICI) therapy.Building on prior data from monocyte-macrophage coculture with fibroblasts, the authors hypothesized a unique role for the combined actions of prostaglandin PGE2 and TNF. The authors studied this combined state using an in vitro model with macrophages derived from monocytes of healthy donors. They complemented this with single-cell transcriptomic and epigenetic data from patients with ICI-RA, specifically, macrophages sorted out of synovial fluid and tissue samples. The study addressed critical questions regarding the regulation of PGE2 and TNF: Are their actions co-regulated or antagonistic? How do they interact with IFN-γ in shaping macrophage responses?This study is the first to specifically investigate a macrophage subset responsive to the PGE2 and TNF combination in the context of ICI-RA, describes a new and easily reproducible in vitro model, and studies the role of IFNgamma regulation of this particular Mф subset.Strengths:Methodological quality: The authors employed a robust combination of approaches, including validation of bulk RNA-seq findings through complementary methods. The methods description is excellent and allows for reproducible research. Importantly, the authors compared their in vitro model with ex vivo single-cell data, demonstrating that their model accurately reflects the molecular mechanisms driving the pathogenicity of this macrophage subset.Weaknesses:Introduction: The introduction lacks a paragraph providing an overview of ICI-induced arthritis pathogenesis and a comparison with other types of arthritis. Including this would help contextualize the study for a broader audience.

Thank you for this suggestion, we have added a paragraph on ICI-arthritis to intro (pg. 4, middle paragraph).

Results Section: At the beginning of the results section, the experimental setup should be described in greater detail to make an easier transition into the results for the reader, rather than relying just on references to Figure 1 captions.

We have clarified the experimental setup (pg. 5).

There is insufficient comparison between single-cell RNA-seq data from ICI-induced arthritis and previously published single-cell RA datasets. Such a comparison may include DEGs and GSEA, pathway analysis comparison for similar subsets of cells. Ideally, an integration with previous datasets with RA-tissue-derived primary monocytes would allow for a direct comparison of subsets and their transcriptomic features.

We thank the Reviewer for this suggestion, which has increased the impact of our data and analysis. A computationally rigorous representation mapping approach showed that ICI-arthritis myeloid subsets predominantly mapped onto 4 previously defined RA subsets including IL-1β+ cells. This result was corroborated using a complementary data integration approach. Analysis of (TNF + PGE)-induced gene sets (TP signatures) in ICI-arthritis myeloid cells projected onto the RA subsets using the AUCell package showed elevated TP gene expression in similar ICI-arthritis and RA monocytic cell subsets. We also found mutually exclusive expression of TP and IFN signatures in distinct RA and ICI-arthritis myeloid cell subsets, which supports that the opposing cross-regulation between IFN-γ and PGE2 pathways that we identified in vitro also functions similarly in vivo. This analysis is shown in the new Fig. 3, described on pg. 7, and discussed on pp. 13-14.

While it's understandable that arthritis samples are limited in numbers and myeloid cell numbers, it would still be interesting to see the results of PGE2+TNF in vitro stimulation on the primary RA or ICI-RA macrophages. It would be valuable to see RNA-Seq signatures of patient cell reactivation in comparison to primary stimulation of healthy donor-derived monocytes.

We agree that this would be interesting but given limited samples and distribution of samples amongst many studies and investigators this is beyond the scope of the current study.

Discussion: Prior single-cell studies of RA and RA macrophage subpopulations from 2019, 2020, 2023 publications deserve more discussion. A thorough comparison with these datasets would place the study in a broader scientific context.Creating an integrated RA myeloid cell atlas that combines ICI-RA data into the RA landscape would be ideal to add value to the field.As one of the next research goals, TNF blockade data in RA and ICI-RA patients would be interesting to add to such an integrated atlas. Combining responders and non-responders to TNF blockade would help to understand patient stratification with the myeloid pathogenic phenotypes. It would be great to read the authors' opinion on this in the Discussion section.

Please see our response to point 3 above. This point is addressed in Fig. 3, pg. 7, and pp. 13-14, which includes a discussion of responders and nonresponders and patient stratification.

Conclusion: The authors demonstrated that while PGE2 maintains the inflammatory profile of macrophages, it also induces a distinct phenotype in simultaneous PGE2 and TNF treatment. The study of this specific subset in single-cell data from ICI-RA patients sheds light on the pathogenic mechanisms underlying this condition, however, how it compares with conventional RA is not clear from the manuscript.Given the substantial incidence of ICI-induced autoimmune arthritis, understanding the unique macrophage subsets involved for future targeting them therapeutically is an important challenge. The findings are significant for immunologists, cancer researchers, and specialists in autoimmune diseases, making the study relevant to a broad scientific audience.
**Reviewer #2 (Public review):**
Summary/Significance of the findings:The authors have done a great job by extensively carrying out transcriptomic and epigenomic analyses in the primary human/mouse monocytes/macrophages to investigate TNF-PGE2 (TP) crosstalk and their regulation by IFN-γ in the Rheumatoid arthritis (RA) synovial macrophages. They proposed that TP induces inflammatory genes via a novel regulatory axis whereby IFN-γ and PGE2 oppose each other to determine the balance between two distinct TNF-induced inflammatory gene expression programs relevant to RA and ICI-arthritis.Strengths:The authors have done a great job on RT-qPCR analysis of gene expression in primary human monocytes stimulated with TNF and showing the selective agonists of PGE2 receptors EP2 and EP4 22 that signal predominantly via cAMP. They have beautifully shown IFN-γ opposes the effects of PGE2 on TNF-induced gene expression. They found that TP signature genes are activated by cooperation of PGE2-induced AP-1, CEBP, and NR4A with TNF-induced NF-κB activity. On the other hand, they found that IFN-γ suppressed induction of AP-1, CEBP, and NR4A activity to ablate induction of IL-1, Notch, and neutrophil chemokine genes but promoted expression of distinct inflammatory genes such as TNF and T cell chemokines like CXCL10 indicating that TP induces inflammatory genes via IFN-γ in the RA and ICI-arthritis.Weaknesses:(1) The authors carried out most of the assays in the monocytes/macrophages. How do APCcells like Dendritic cells behave with respect to this TP treatment similar dosing?

We agree that this is an interesting topic especially as TNF + PGE2 is one of the standard methods of maturing in vitro generated human DCs and promoting antigen-presenting function. As DC maturation is quite different from monocyte activation this would represent a new study and is beyond the scope of the current manuscript. We have instead added a paragraph to the discussion (pg. 12) and cited the literature on DC maturation by TNF + PGE2 including one of our older papers (PMID: 18678606; 2008)

(2) The authors studied 3h and 24h post-treatment transcriptomic and epigenomic. What happens to TP induce inflammatory genes post-treatment 12h, 36h, 48h, 72h. It is critical to see the upregulated/downregulated genes get normalised or stay the same throughout the innate immune response.

We now clarify that subsets of inducible genes showed distinct kinetics of induction with transient expression at 3 hr versus sustained expression over the 24 hr stimulation period as shown in Supplementary Fig. 1 (pg. 5).

(3) The authors showed IL1-axis in response to the TP-treatment. Do other cytokine axes get modulated? If yes, then how do they cooperate to reduce/induce inflammatory responses along this proposed axis?

This is an interesting question, which we approached using a combination of pathway analysis and targeted inspection of pathways important pathogenesis of RA, which is the inflammatory condition most relevant for this study. In addition to genes in the IL-1-NF-κB core inflammatory pathway, pathway analysis of genes induced by TP co-stimulation showed enrichment of genes related to leukocyte chemotaxis, in particular neutrophil migration. Accordingly, TP costimulation increased expression of CSF3, which plays a key role in mobilizing neutrophils from the bone marrow, and major neutrophil chemokines CXCL1, CXCL2, CXCL3 and CXCL5 that recruit neutrophils to sites of inflammation including in inflammatory arthritis. Analysis of the late response to TNF similarly showed enrichment of genes important in chemotaxis, and suppression of genes in the cholesterol biosynthetic pathway, which we and others have previously linked to IFN responses. Targeted inspection of genes in additional pathways implicated in RA pathogenesis showed increased expression of genes in the Notch pathway. We believe that these pathways work together with the IL-1 pathway to increase immune cell recruitment and activation in inflammatory responses; these results are described on pp. 5-6 and are incorporated into Figures 1, 2 and Supplementary Fig. 2.

Overall, the data looks good and acceptable but I need to confirm the above-mentioned criticisms.

**Recommendations for the authors:**

**Reviewer #1 (Recommendations for the authors):**
The discussion section of the manuscript claims: "In this study, we utilized transcriptomics to demonstrate a 'TNF + PGE2' (TP) signature in RA and ICI-arthritis IL-1β+ synovial macrophages." This statement is misleading, as no new transcriptomic data from RA synovial samples were generated in this study. To support such a claim, the authors would need to compare primary monocytes or macrophages from RA patients using bulk RNA-seq or singlecell RNA-seq. Based on the current data, the comparison is limited to bulk RNA-seq findings from the authors' in vitro model and prior monocyte-fibroblast coculture studies.

We have modified the abstract and discussion (pg. 10) to reflect that we have compared an in vitro generated TP signature with gene expression in previously identified RA macrophage subsets.